# Hedgehog signaling via Gli2 prevents obesity induced by high-fat diet in adult mice

Yu Shi[1], Fanxin Long[1,2,3]*

[1]Department of Orthopaedic Surgery, Washington University School of Medicine, St. Louis, United States; [2]Department of Medicine, Washington University School of Medicine, St. Louis, United States; [3]Department of Developmental Biology, Washington University School of Medicine, St. Louis, United States

**Abstract** Obesity poses a significant risk of developing type II diabetes and other diseases. Hedgehog (Hh) signaling has been shown to inhibit adipose tissue development, but its effect on diet-induced obesity during postnatal life is not known. Here by inducing expression of constitutively active Smoothened (SmoM2) or Gli2 (ΔNGli2) in the adipocyte lineage of postnatal mice, we show that targeted activation of Hh signaling suppresses high-fat-diet-induced obesity and improves whole-body glucose tolerance and insulin sensitivity. Both SmoM2 and ΔNGli2 induce the expression of Wnt6, a known anti-adipogenic factor, in fat depots of the mouse. Hh-Gli2 signaling inhibits not only adipocyte differentiation but also lipogenesis in adipocytes in vitro. Finally, pharmacological inhibition of Porcupine, an acyltransferase essential for Wnt secretion, alleviates both anti-adipogenic and anti-lipogenic effects of Hh in cell culture models. Overall, targeted activation of Hh signaling ameliorates diet-induced obesity and may be explored for pharmaceutical development.

DOI: https://doi.org/10.7554/eLife.31649.001

*For correspondence:
flong@wustl.edu

Competing interests: The authors declare that no competing interests exist.

## Introduction

The global epidemic of obesity affects hundreds of millions of people worldwide in 2016, as estimated by the World Health Organization. In the United States, one third of the adult population is obese (*Flegal et al., 2002*). It is well known that clinically obese individuals exhibit a markedly higher chance of developing cardiovascular disease, type II diabetes, cancer and stroke. A recent meta-analysis shows that obesity is associated with significantly higher all-cause mortality (*Flegal et al., 2013*). Although the obesity epidemic clearly requires comprehensive solutions, pharmacological strategies are urgently needed.

Hedgehog (Hh) signaling is an evolutionarily conserved pathway controlling tissue development and homeostasis. In this pathway, Hh ligands bind to the receptor Patched 1 (Ptch1) to relieve its inhibition on Smoothened (Smo), a seven-pass transmembrane protein, resulting in transcriptional activation by the Gli family of transcription factors (*Ingham and McMahon, 2001*; *Goetz et al., 2009*). However, point mutations in Smoothened, such as W535L (hereafter SmoM2) originally discovered in human sporadic basal-cell carcinoma, can activate Gli-mediated transcription independent of a Hh ligand (*Xie et al., 1998*; *Long et al., 2001*; *Jeong et al., 2004a*). Of the three Gli proteins in mammals, Gli2 and Gli3 are the primary effectors to transduce Hh signaling whereas Gli1, a direct target of Gli2 and Gli3, functions to amplify the transcriptional response of Hh signaling (*Bai et al., 2002*; *Park et al., 2000*). Moreover, Gli2 is predominantly a transcriptional activator in response to Hh whereas Gli3 mainly exists as a repressor that is de-repressed upon Hh signaling (*Wang et al., 2000*; *Pan et al., 2006*; *Bai and Joyner, 2001*; *Hilton et al., 2005*). The activator

function of Gli2 normally requires Hh input but an N-terminally truncated form (ΔNGli2) has been found to stimulate transcription constitutively (*Sasaki et al., 1999*; *Joeng and Long, 2009*; *Mill et al., 2003*).

Hh signaling has been implicated in the development of adipose tissues. Genetic studies in Drosophila identified Hh as a potent inhibitor of fat body formation (*Pospisilik et al., 2010*; *Suh et al., 2006*). Deletion of Sufu, an endogenous inhibitor of Hh signaling, with aP2-Cre impaired the formation of white (WAT) but not brown (BAT) adipose tissue in mice (*Pospisilik et al., 2010*). However, a recent study showed that Hh activation through deletion of Ptch1 or expression of SmoM2 with aP2-Cre inhibited BAT development in newborn mice (*Nosavanh et al., 2015*). Because all of the mouse genetic studies to date perturbed Hh signaling throughout embryogenesis, it is not known whether Hh signaling can influence adiposity when activated specifically at the adult stage.

Besides Hh, Wnt proteins have also been shown to inhibit adipogenesis. Transgenic mice expressing Wnt10b from the *Fabp4* (also known as *aP2*) promoter resisted fat accumulation (*Longo et al., 2004*). In vitro studies have also implicated Wnt6 and Wnt10a in the suppression of adipocyte differentiation (*Cawthorn et al., 2012*). Although Wnt acts downstream of Hh in the context of osteoblast differentiation, it is not known whether or how the two signals interact during fat formation (*Hu et al., 2005*).

In the present study, by inducing the expression of either SmoM2 or ΔNGli2 in the adipocyte lineage of postnatal mice, we show that Hh signaling exerts a relatively modest effect in mice on the regular diet, but notably suppresses both WAT and BAT accumulation caused by a high-fat diet. We further demonstrate that Hh induces Wnt expression to inhibit not only adipocyte differentiation but also the conversion of glucose to lipids.

## Results

### Hedgehog activation suppresses high-fat-diet-induced obesity and metabolic symptoms

To activate Hh signaling in the adipose tissue specifically in postnatal mice, we wished to use the Pparg-tTA allele to target the adipocyte lineage following withdrawal of doxycycline (Dox) from the drinking water. To this end, we first assessed the tissue specificity and efficacy of Pparg-tTA by generating mice with the genotype of Pparg-tTA;TetO-Cre;mT/mG and monitoring GFP expression in response to Dox withdrawal. After two months of Dox withdrawal starting at two months of age, we observed GFP in both the gonadal fat (white adipose tissue, hereafter WAT) and the interscapular fat depot (brown adipose tissue, hereafter BAT) as well as the bone marrow fat, but not in the liver, the intestine or the heart (*Figure 1A*). However, immunostaining indicated that only a subset of perilipin +adipocytes expressed GFP (48 ± 6.1% in WAT; 35 ± 4.1% in BAT; 18 ± 3.0% in bone marrow fat, n = 3), indicating mosaic Cre activity due to varied expression of tTA or uneven clearance of Dox, or both. Thus, Pparg-tTA in combination with TetO-Cre predominantly but incompletely targets the adipose tissue in adult mice after Dox withdrawal.

We next used Pparg-tTA to activate Hh signaling in the adipocyte lineage in postnatal mice. Specifically, we raised mice harboring one allele each of Pparg-tTA, TetO-Cre and R26-SmoM2 (genotype Pparg-tTA;TetO-Cre;R26$^{SmoM2/+}$) on regular chow plus Dox from conception through two months of age, at which point they either continued on Dox (+Dox) or were weaned off Dox (-Dox) for up to 8, 18 or 26 weeks (*Figure 1B*). After 8 weeks of Dox withdrawal, the Hh target genes Ptch1 and Gli1 were markedly induced whereas the adipocyte marker genes Pparg, Cebpa and Fabp4 were suppressed in WAT, thus confirming the efficacy of the approach (*Figure 1C*). However, the mice maintained a normal body weight even after 18 weeks of Dox withdrawal although they eventually showed a decrease after 26 weeks (*Figure 1D*). At 26 weeks, the –Dox mice also exhibited a lower percentage of body fat but glucose metabolism appeared to be normal according to glucose tolerance tests (GTT) (*Figure 1E,F*). Thus, prolonged Hh activation in the adipose tissue suppresses fat accumulation without altering glucose homeostasis in mice fed with the regular chow.

We next examined whether Hh activation affects obesity caused by a high fat diet. For this, the triple transgenic mice (Pparg-tTA;TetO-Cre;R26$^{SmoM2/+}$) raised on regular chow and Dox from conception through two months of age were subjected to a high-fat diet (HFD) for up to 16

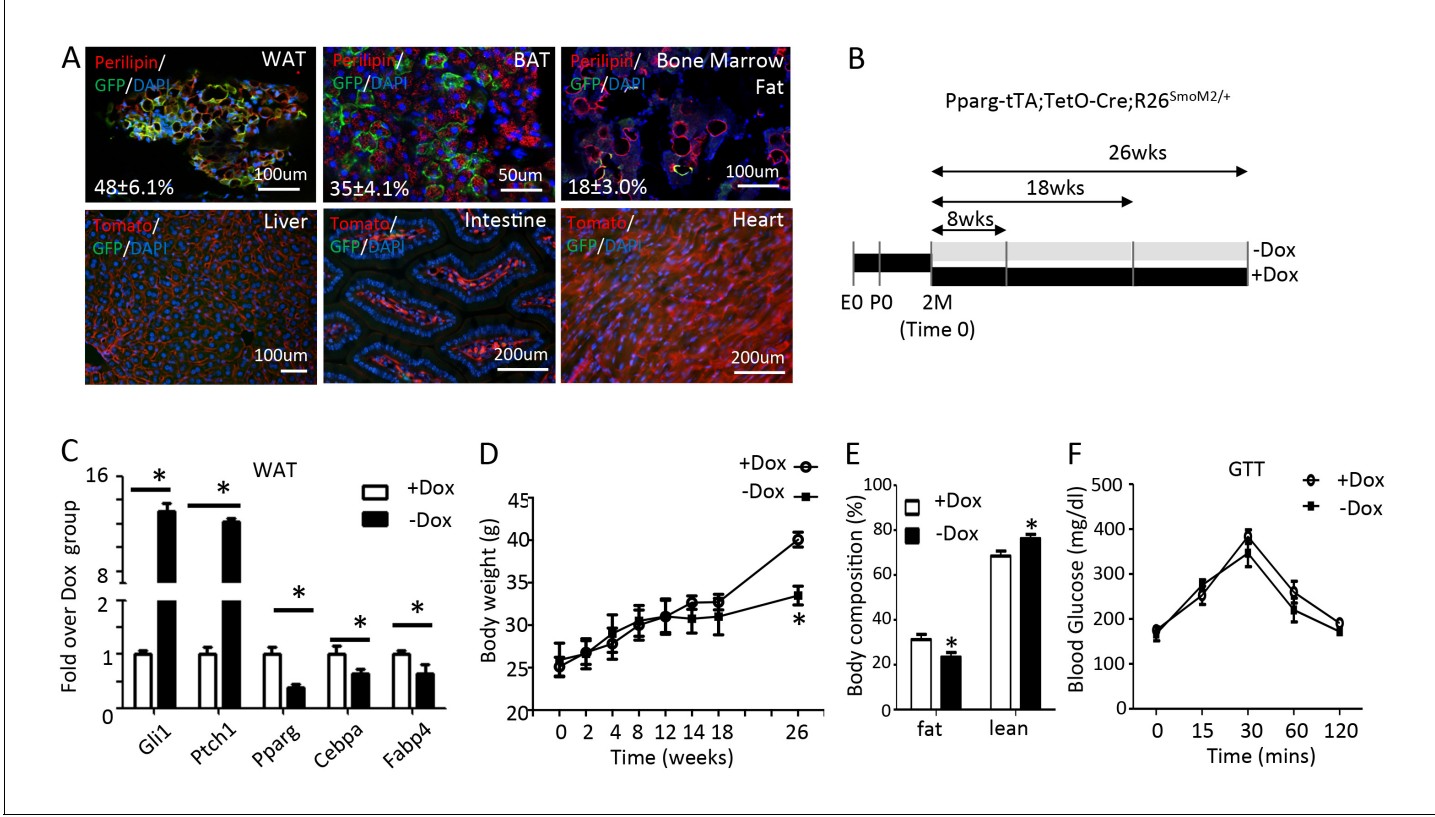

**Figure 1.** Long-term activation of Smo reduces fat accumulation without altering glucose metabolism in mice on regular chow diet. (**A**) Pparg-tTA targets adipose tissues in adult mice. Pparg-tTA;TetO-Cre;mT/mG mice were analyzed after two months of Dox withdrawal starting at two months of age. GFP and perilipin were detected by immunofluorescence staining whereas tdTomato fluorescence was visualized directly on frozen sections of adipose tissues. Percentage of adipocyte targeted indicated for each fat depot. DAPI stains DNA. WAT: white adipose tissue (gonadal fat); BAT: brown adipose tissue (interscapular fat). (**B**) Experimental design for activating Smo in adipose tissue of adult mice. E0: embryonic day 0; P0: postnatal day 0; 2M: 2 months of age. (**C**) Expression of adipogenic genes in WAT after 8 weeks of Dox regimen. (**D**) Measurements of body weights at different time points of Dox regimen. (**D, E**) Body composition (**D**) and GTT (**E**) after 26 weeks of Dox. *p<0.05, n = 5 mice, males. Females show similar results.
DOI: https://doi.org/10.7554/eLife.31649.002

weeks, with or without continued Dox treatment (+Dox or –Dox, respectively) (*Figure 2A*). The –Dox mice were noticeably leaner than +Dox group after 8 weeks of HFD treatment (*Figure 2B, C*). At 8 weeks, the –Dox mice already showed a lower percentage of fat than the +Dox mice (*Figure 2D*). Both the gonadal (WAT) and the interscapular (BAT) fat depots were markedly reduced in the -Dox mice, consistent with a notable decrease in the size of adipocytes (*Figure 2E,F*). In addition, the bone marrow adipocytes, readily detectable by perilipin immunostaining in the long bones of the +Dox mice, were essentially absent in the –Dox group (*Figure 2G*). Molecular analyses confirmed that the adipocyte marker genes were significantly suppressed in WAT and BAT of the –Dox mice, whereas Gli1 and Ptch1 were greatly elevated (*Figure 2H,I*). Moreover, the induction of Gli1 was restricted to WAT and BAT but not the other tissues in the -Dox mice, confirming the intended specificity to adipose tissues (*Figure 2J*). Finally, because obesity often leads to glucose intolerance and insulin resistance, we examined whether suppression of fat accumulation by Hh results in metabolic benefits. The –Dox mice exhibited faster clearance of glucose in the glucose tolerance test (GTT), and greater sensitivity to insulin in the insulin tolerance test (ITT) than their +Dox counterparts (*Figure 2K,L*). To test the possibility that the phenotypes here might be due to the lipodystrophic effect of the Pparg-tTA allele or the antimicrobial effect of Dox as previously reported, we repeated the experiment with mice carrying either no transgene or only Pparg-tTA (*Kim et al., 2007*; *Cho et al., 2012*). After 8 weeks of HFD with or without Dox (starting at two months of age), we did not observe any difference in

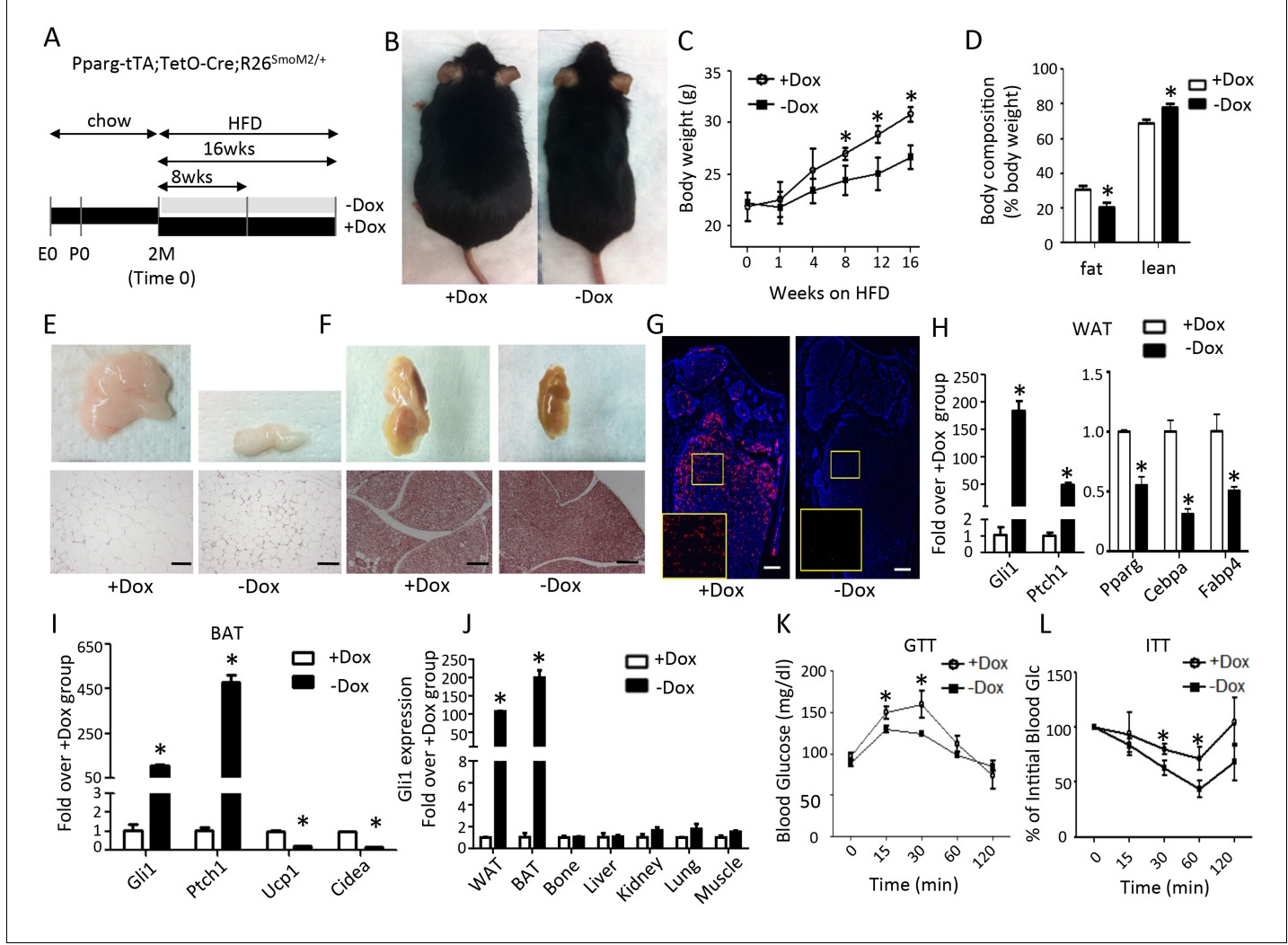

**Figure 2.** Smo activation prevents obesity and improves glucose metabolism in mice on high fat diet. (A) A schematic for experimental design. (B) Representative images after 8 weeks on HFD. (C) Measurements of body weight. (D) Body composition after 8 weeks on HFD. (E, F) Whole-mount images (upper) and histology (lower) of gonadal white fat (E) or interscapular brown fat (F) after 8 weeks on HFD. (G) Detection of bone marrow fat by perilipin immunofluorescence staining after 8 weeks on HFD. Boxed regions shown at higher magnification in insets. (H, I) Gene expression by qPCR in WAT (H) and BAT (I) after 8 weeks on HFD. (J) Gli1 expression in different tissues after 8 weeks on HFD. (K, L) Glucose tolerance test (GTT) (K) and insulin tolerance test (ITT) (L) after 8 weeks on HFD. *p<0.05, n = 5 mice, females. Males show similar results. Black scale bar: 100 μm; white scale bar: 200 μm.

DOI: https://doi.org/10.7554/eLife.31649.003

The following figure supplements are available for figure 2:

**Figure supplement 1.** No obvious effect for Dox or Pparg-tTA alone on whole body metabolism.
DOI: https://doi.org/10.7554/eLife.31649.004
**Figure supplement 2.** Characterization of adipogenesis in M2 cells.
DOI: https://doi.org/10.7554/eLife.31649.005
**Figure supplement 3.** Hh signaling suppresses adipogenesis in M2 cells.
DOI: https://doi.org/10.7554/eLife.31649.006

body weight, body composition or glucose metabolism between the +Dox and –Dox groups (*Figure 2—figure supplement 1*). Thus, Hh activation in the Pparg-lineage is sufficient to suppress obesity and improve glucose metabolism in response to a high-fat diet.

## Hedgehog inhibits adipogenesis via Gli2

Although Hh signaling has been shown previously to inhibit adipogenesis, the underlying mechanism is not fully understood. To gain additional insights, we studied the anti-adipogenenic effect of Hh signaling in the murine bone marrow mesenchymal progenitor cell line M2-10B4 (hereafter M2), which we have previously shown to respond robustly to Hh signaling (*Shi et al., 2015a*). The M2 cells underwent adipogenesis when cultured in the adipogenic medium, as indicated by the induction of Cebpb and Cebpd as early as 6 hr and that of Pparg, Cebpa and Fabp4 after 48 hr (*Figure 2—figure supplement 2*). Addition of the Hh agonist purmorphamine (PM) to the adipogenic media completely abolished adipogenesis, as indicated by the loss of oil-red O staining (*Figure 2—figure supplement 3*). Interestingly, PM markedly suppressed Pparg, Cebpa and Fabp4, but not Cebpb or Cebpd that is known to function at an earlier stage of adipogenesis (*Figure 2—figure supplement 3*). Thus, Hh signaling inhibits adipocyte differentiation in a stage-specific manner.

We next sought to distinguish the relative contribution of the different Gli transcription factors to the anti-adipogenic function of Hh. Knockdown of Gli2 with shRNA reduced the mRNA level of Gli2 by 75%, and essentially nullified the inhibitory effect of PM on Pparg, Cebpa and Fabp4 induction by the adipogenic media (*Figure 3A*). In contrast, knocking down either Gli1 or Gli3 to a similar degree did not blunt the anti-adipogenic effect of PM (*Figure 3—figure supplement 1*). Oil red O staining confirmed that knockdown of Gli2 but not Gli1 or Gli3 completely restored the number of adipocytes in the presence of PM (*Figure 3B*). To confirm the anti-adipogenic effect of Gli2, we cultured mouse embryonic fibroblasts (MEF) from the R26$^{\Delta NGli2/+}$ mice, and activated expression of ΔNGli2 (a constitutively active form of Gli2) from the Rosa26 locus with an adenovirus expressing Cre (Ad-Cre) (*Joeng and Long, 2009*). ΔNGli2 essentially abolished the induction of Cebpa, Pparg and Fabp4 mRNA as well as the oil red O-positive cells by the adipogenic media (*Figure 3C,D*). Thus, Gli2 is the principal mediator for Hh to inhibit adipogenesis.

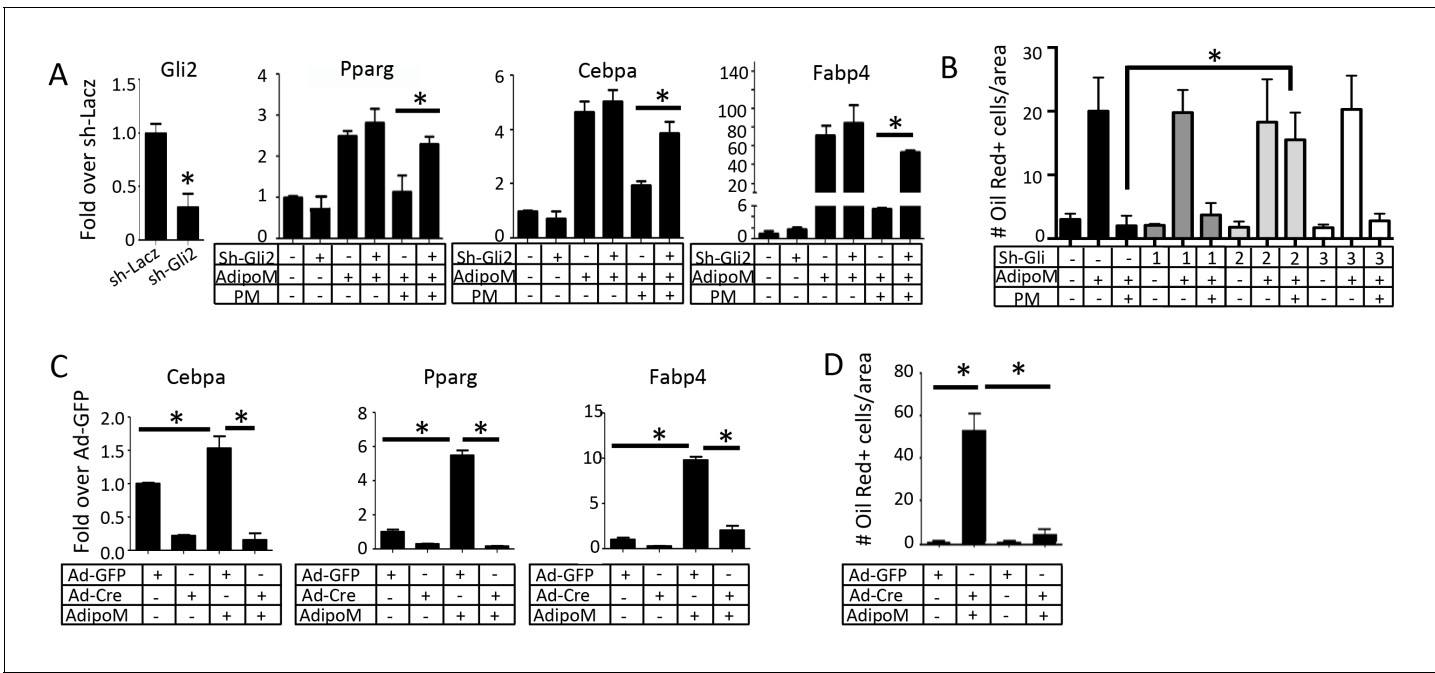

**Figure 3.** Gli2 mediates Hh inhibition of adipogenesis. (**A**) Effects of Gli2 knockdown on the suppression of adipocyte marker genes by purmorphamine (PM). qPCR data normalized to 18S rRNA. (**B**) Effects of Gli1-3 knockdown on oil red O staining. Quantification was shown for number of positive cells per area. (**C**) Expression of adipocyte marker genes in R26$^{\Delta NGli2/+}$ MEF infected with Ad-Cre or Ad-GFP. (**D**) Quantification of oil red O staining in R26$^{\Delta NGli2/+}$ MEF infected with Ad-Cre or Ad-GFP. AdipoM: adipogenic medium. *p<0.05, n = 3.

DOI: https://doi.org/10.7554/eLife.31649.007

The following figure supplement is available for figure 3:

**Figure supplement 1.** Knockdown of either Gli1 or Gli3 does not blunt the anti-adipogenic effect of PM.
DOI: https://doi.org/10.7554/eLife.31649.008

## Constitutively active Gli2 prevents high fat diet-induced obesity

The data so far indicate that Hh signaling suppresses adipogenesis via Gli2 activation. This finding predicts that Gli2 activation would recapitulate the effect of Hh signaling on fat accumulation in vivo. To test this prediction, we used the same Dox regimen as described above to express ΔNGli2 in the adipocyte lineage (*Figure 4A*). Briefly, mice with the genotype of Pparg-tTA;TetO-cre;R26^ΔNGli2/+ were maintained on Dox and regular chow from conception to two months of age before being separated into two groups, with both on HFD but one continuing on Dox (+Dox) and the other off dox (-Dox) for up to 18 weeks. The –Dox mice was significantly leaner than their +Dox counterparts after 8 weeks of HFD, and their difference in body weight increased with time (*Figure 4B,C*). Body composition analyses with MRI revealed a notable reduction in fat and a corresponding increase in lean mass in the –Dox mice after 8 weeks (*Figure 4D*). At that time, both gonadal (WAT) and interscapular fat depots (BAT) were diminished and contained smaller adipocytes in the –Dox mice (*Figure 4E,*

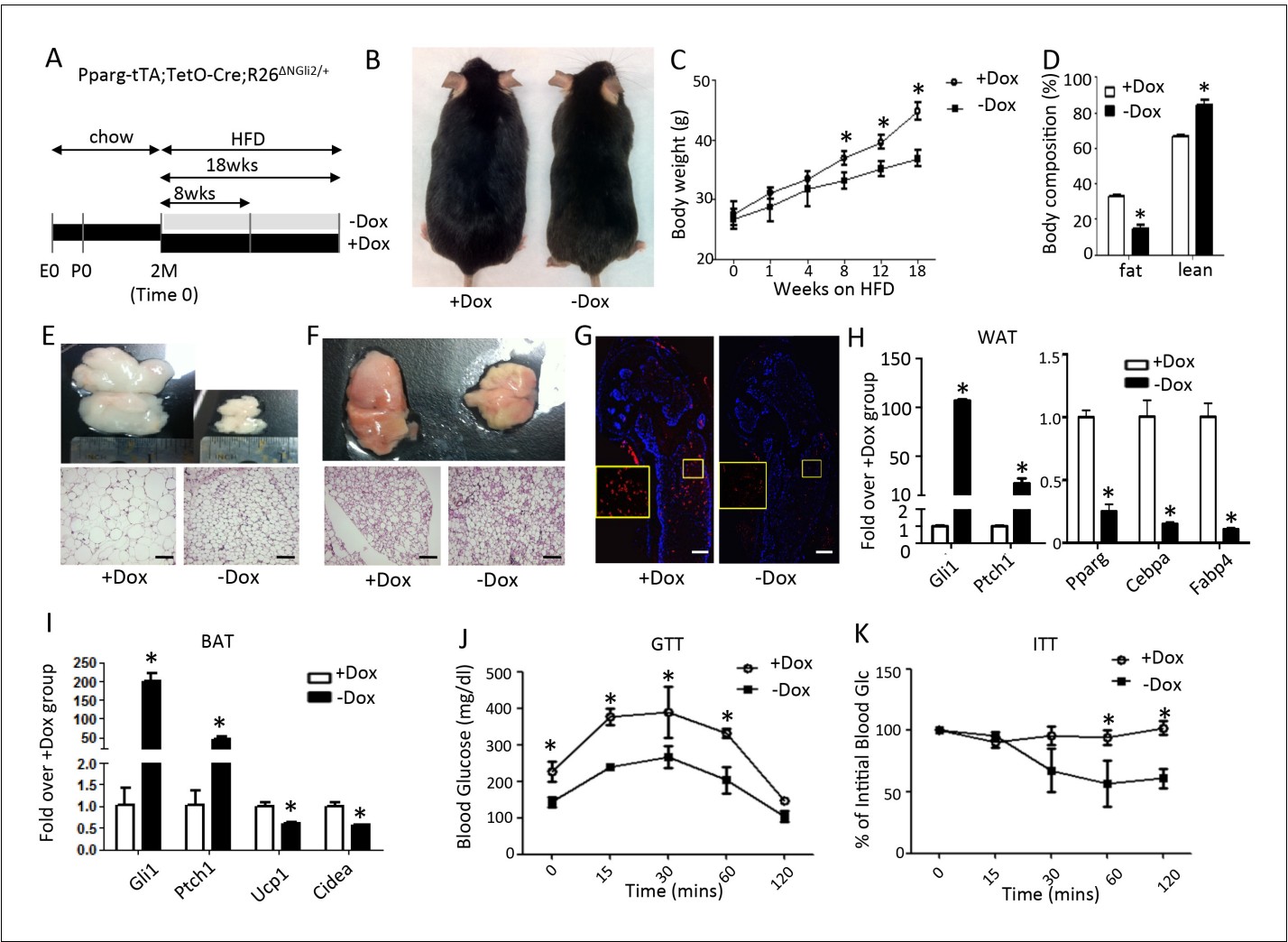

**Figure 4.** Constitutively active Gli2 prevents HFD-induced obesity and improves glucose metabolism. (**A**) A schematic for experimental design. E0: embryonic day 0; P0: postnatal day 0; 2M: 2 months of age. (**B**) Representative images after 8 weeks of HFD. (**C**) Measurements of body weight. (**D**) Measurements of body composition after 8 weeks of HFD. (**E, F**) Whole-mount images (upper) and histology (lower) of gonadal white fat (**E**) or interscapular brown fat (**F**) after 8 weeks of HFD. (**G**) Detection of bone marrow fat by perilipin immunofluorescence staining after 8 weeks on HFD. Boxed regions shown at higher magnification in insets. (**H, I**) Detection of gene expression by qPCR in WAT (**G**) and BAT (**H**) after 8 weeks on HFD. (**J, K**) Glucose tolerance test (GTT) (**I**) and insulin tolerance test (ITT) (**J**) after 8 weeks on HFD. *p<0.05, n = 5 mice, males. Females show similar results. Black scale bar: 100 μm; white scale bar: 200 μm.

DOI: https://doi.org/10.7554/eLife.31649.009

*F*). Immunostaining of perilipin on long bone sections revealed that the bone marrow fat was essentially eliminated in the –Dox mice (*Figure 4G*). Molecular analyses confirmed that Dox withdrawal led to marked induction of Ptch1 and Gli1 in both WAT and BAT, reduction of Pparg, Cebpa, Fabp4 in WAT, and suppression of Ucp1, Cidea in BAT (*Figure 4H,I*). Finally, compared to the +Dox counterparts, the –Dox mice exhibited a lower basal glucose level, better glucose tolerance and greater insulin sensitivity after 8 weeks of HFD treatment (*Figure 4J,K*). Thus, like SmoM2, postnatal activation of Gli2 in the adipocyte lineage suppresses obesity and metabolic dysfunction caused by a high fat diet.

## Wnt6 is induced by Hh-Gli2 signaling

We next searched for potential downstream mediators for the anti-adipogenic function of Hh signaling. We have previously performed RNA-seq to compare the mRNA expression profile in M2 cells with or without purmorphamine (PM) for 72 hr (*Shi et al., 2015b*). Those experiments revealed approximately 750 genes exhibiting at least a 2-fold change (365 up and 382 down) in response to Hh activation (*Supplementary file 1*). Among the upregulated genes, Wnt5a, Wnt6 and Wnt9a attracted our attention as Wnt signaling is known to inhibit adipogenesis. RT-qPCR confirmed the induction of all three Wnt genes by PM in M2 cells, with that of Wnt6 being most robust, reaching over 10 fold after 48 hr (*Figure 5A*). Knockdown of Gli2 with shRNA essentially eliminated the induction of the Wnt genes by PM (*Figure 5B*). Consistent with the findings in M2 cells, MEFs isolated from the R26$^{\Delta NGli2/+}$ mouse and infected with Ad-Cre upregulated Wnt5a, Wnt6 and Wnt9a expression by 216, 1200 and 64 fold, respectively, over the cells infected with Ad-GFP (*Figure 5C*). Moreover, when RNA from the whole gonadal fat pad was analyzed, Wnt6 was induced when either

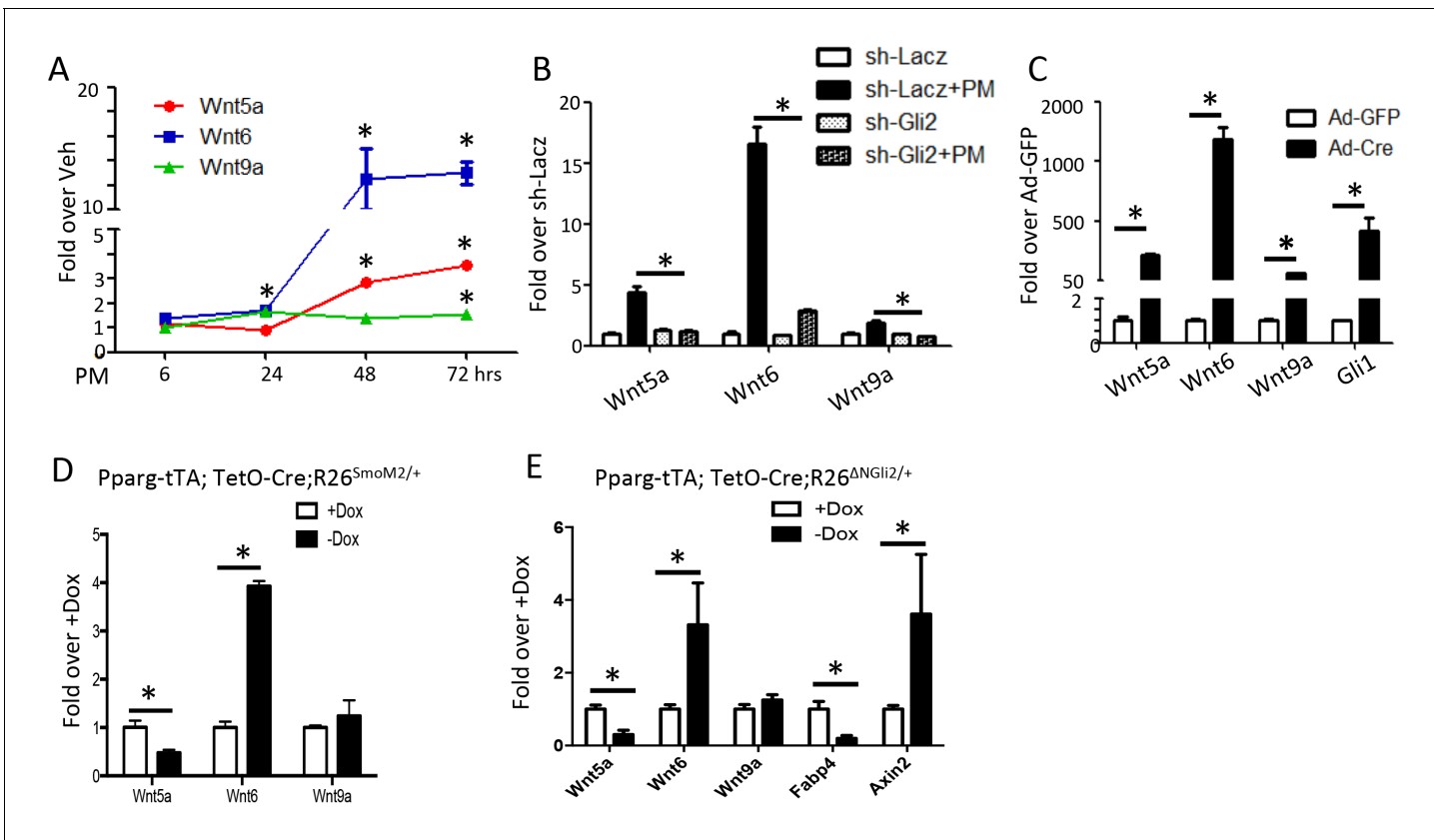

**Figure 5.** Wnt6 is a potential target of Gli2. (A–E) qPCR analyses in M2 cells with PM treatment (A), in M2 cells with PM treatment and shRNA knockdown (B), in R26$^{\Delta NGli2/+}$ MEF cells infected with Ad-GFP or Ad-Cre (C), in gonadal fat pat isolated from Pparg-tTA;TetO-Cre;R26$^{SmoM2/+}$ (D) or Pparg-tTA;TetO-Cre;R26$^{\Delta NGli2/+}$ mice (E) after 8 weeks of HFD with or without Dox. 18S rRNA was used as the internal control for all qPCR analyses. *p<0.05, n = 3.

DOI: https://doi.org/10.7554/eLife.31649.010

SmoM2 or ΔNGli2 was activated upon Dox withdrawal, although Wnt5a or Wnt9a was either reduced or unchanged (*Figure 5D,E*). The induction of Wnt6 could be relevant as it was previously shown to inhibit adipogenesis (*Cawthorn et al., 2012*). Importantly, Axin2, a prototypic Wnt target gene, was upregulated in the gonadal fat pad upon ΔNGli2 expression, confirming activation of Wnt/β-catenin signaling in vivo (*Figure 5E*). The failure to detect Wnt5a or Wnt9b induction in the whole fat depot may result from the cellular heterogeneity, and the incomplete penetrance of Pparg-tTA within the tissue as shown earlier. Alternatively, Wnt5a or Wnt9b may not be induced by Hh signaling in vivo. Overall, Hh activation induces Wnt6 expression and likely activates β-catenin signaling in the adipose tissue, but the specific contribution of Wnt6 to the anti-adipogenic activity of Hh remains to be tested in vivo.

## Inhibition of Wnt secretion relieves Hh anti-adipogenic function in vitro

We then tested whether Hh requires de novo Wnt production to inhibit adipogenesis in vitro. Because palmitoylation by O-acyltransferase Porcupine (Porcn) is essential for Wnt secretion, we used IWP2, a small molecule inhibitor of Porcn, to inhibit paracrine Wnt signaling (*Chen et al., 2009*). In M2 cells, IWP2 reduced the mRNA level of Nkd2, a known transcriptional target of Wnt-β-catenin signaling, confirming the efficacy of the inhibitor (*Figure 6A*). In the adipogenic media, IWP2 modestly stimulated the expression of Pparg and Fabp4, but greatly relieved the suppression of Pparg, Cebpa and Fabp4 by PM (*Figure 6B*). Quantification of adipocytes with oil red O staining confirmed that IWP2 significantly restored adipogenesis in the face of PM (*Figure 6C*). To test the relationship between Wnt and Hh signaling in primary cells, we isolated preadipocytes from the gonadal fat pad of two-month-old Pparg-tTA;TetO-Cre;R26$^{\Delta NGli2/+}$ mice that had been maintained on Dox since conception (no ΔNGli2 expression), and then cultured the cells for 3 days in either growth media or adipogenic media with (+Dox) or without Dox (-Dox). As expected, the –Dox cells expressed more Gli1 mRNA but responded significantly less to the adipogenic stimuli that induce Pparg and Fabp4 (*Figure 6D*). However, when IWP2 was added to the adipogenic media, the induction of Pparg and Fabp4 was improved in the –Dox cells (*Figure 6E*). Thus, inhibition of Wnt secretion partially relieves the suppression of adipogenesis by Hh signaling.

## Hedgehog signaling reduces glucose contribution to lipid in adipocytes

Overexpression of either SmoM2 or ΔNGli2 resulted in a smaller size of adipocytes. This finding indicates that Hh signaling suppresses adipocyte hypertrophy in response to high fat diet. To examine this regulation in more detail, we induced primary preadipocytes from wild type mice to form adipocytes (three days in adipogenic media) and then assessed the effect of PM specifically on the lipid-accumulating phase (eleven days in insulin-only media). RT-qPCR assays indicated that PM significantly reduced the expression of lipogenesis genes such as Lpl (lipoprotein lipase), Fasn (fatty acid synthase), Plin (perilipin) and Dgat1 (diacylglycerol acyltransferases 1) without affecting the differentiation marker Fabp4 (*Figure 7A*). Furthermore, by measuring the size of the oil-red-O-stained lipid droplets, we found that PM significantly reduced the average droplet size (*Figure 7B*). To assess the relevance of Wnt secretion in this regulation, we tested the effect of IWP2 specifically on the lipid-accumulating phase (eleven days in insulin-containing media) either alone or together with PM. IWP2 alone did not have an obvious effect on the size of lipid droplets, but it eliminated the suppression by PM (*Figure 7B*). To determine the role of Gli2 in lipid formation, we performed similar experiments with primary preadipocytes isolated from R26$^{\Delta NGli2/+}$ mice. Specifically, we induced adipocyte differentiation for three days and then activated ΔNGli2 expression with Ad-Cre in the insulin-only media with or without IWP2 for eleven days. Whereas expression of ΔNGli2 induced by Ad-Cre reduced the average size of lipid droplets, IWP2 abolished the effect (*Figure 7C*). Thus, besides inhibiting adipocyte differentiation, Hh-Gli2 signaling suppresses lipid accumulation in adipocytes through a Wnt-mediated mechanism.

Because glucose is a major carbon source for lipid formation, we sought to determine the effect of Hh signaling on glucose utilization by adipocytes. After primary preadipocytes from wild-type mice were induced for three days to form adipocytes, glucose consumption was determined for the next three days when lipid accumulated in response to insulin with or without PM or IWP2. PM significantly decreased glucose consumption and this reduction was rescued by IWP2 (*Figure 7D*). However, neither PM nor IWP2 had any effect on lactate levels in the media, indicating that the decrease

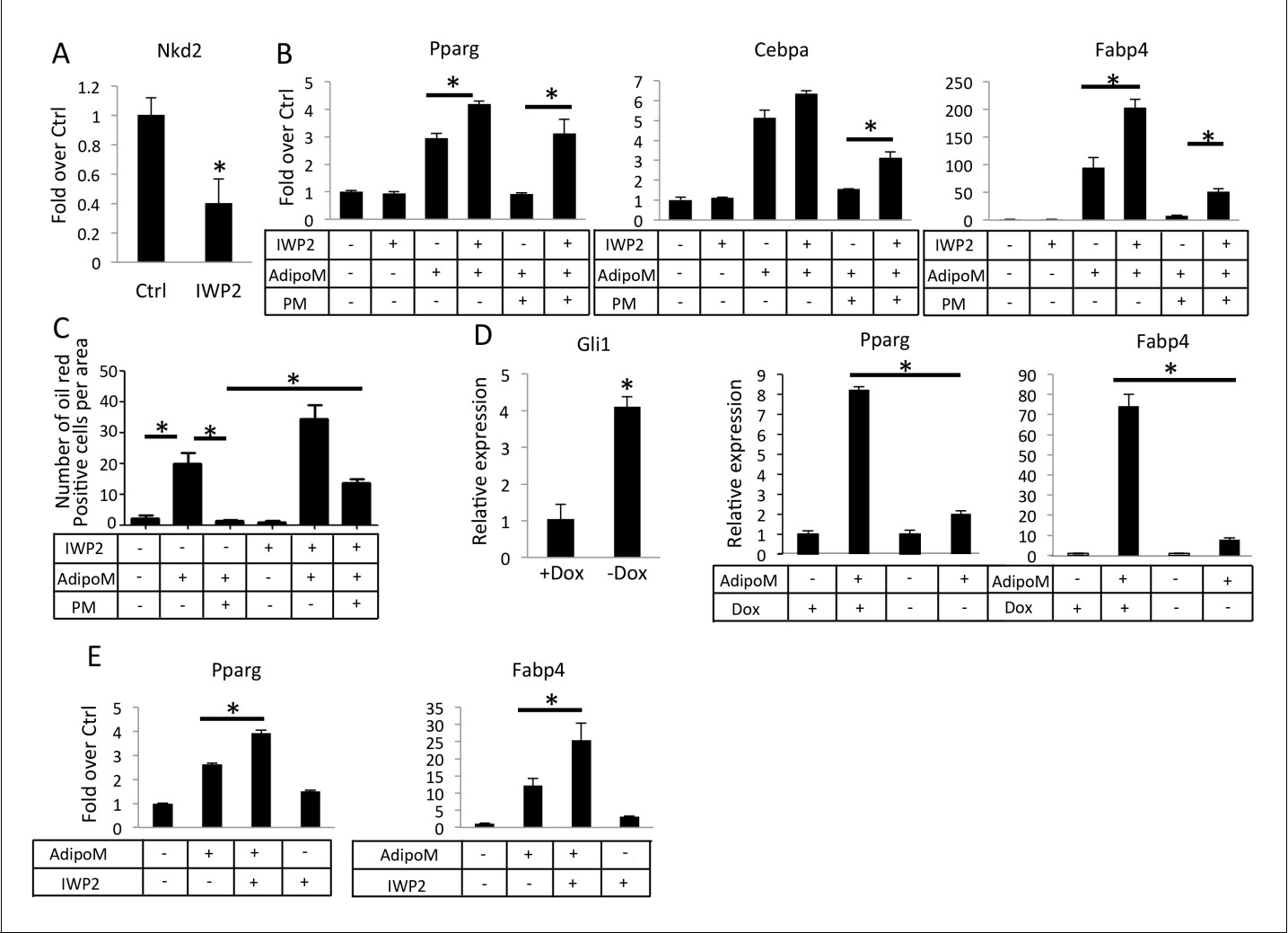

**Figure 6.** Inhibition of Wnt secretion ameliorates Hh suppression of adipogenesis. (**A**) IWP2 reduces Nkd2 mRNA levels in M2 cells. (**B**) IWP2 partially rescued suppression of adipocyte gene expression by PM in M2 cells. (**C**) IWP2 partially rescued the number of oil red O positive cells suppressed by PM. (**D**) Expression of ΔNGli2 (-Dox) induced Gli1 but suppressed adipocyte marker genes in preadipocytes cultured from PPARgtTA;TetOcre; R26^ΔNGli2/+ mice. (**E**) IWP2 partially rescued adipogenic differentiation in preadipocytes isolated from PPARgtTA;TetOcre;R26^ΔNGli2/+ mice and cultured without Dox. 18S rRNA was used for normalization in all qPCR analyses. *p<0.05, n = 3.
DOI: https://doi.org/10.7554/eLife.31649.011

in glucose consumption in response to Hh signaling likely reduced the glucose flux to other fates such as lipid formation (*Figure 7E*). To investigate directly the contribution of glucose to lipid formation, we tracked the incorporation of glucose carbons to cellular lipids by adding the radioactively labeled [U-$^{14}$C6]-glucose to the insulin-only media for the final three days of the lipid-accumulating phase (total 11 days) with or without the addition of PM or IWP2. PM significantly decreased the $^{14}$C contribution to lipid whereas IWP2 had the opposite effect. Importantly, IWP2 restored glucose contribution to the control level in the presence of PM even though the rescued level was lower than that by IWP2 alone (*Figure 7F*). Taken together, the results demonstrate that Hh signaling functions partly through Wnt induction to inhibit not only adipocyte differentiation but also lipid accumulation in adipocytes (*Figure 7G*).

## Discussion

We report that activation of Hh signaling is effective in suppressing obesity and the associated metabolic abnormalities caused by high fat diet in adult mice. At the cellular level, Hh inhibits not only

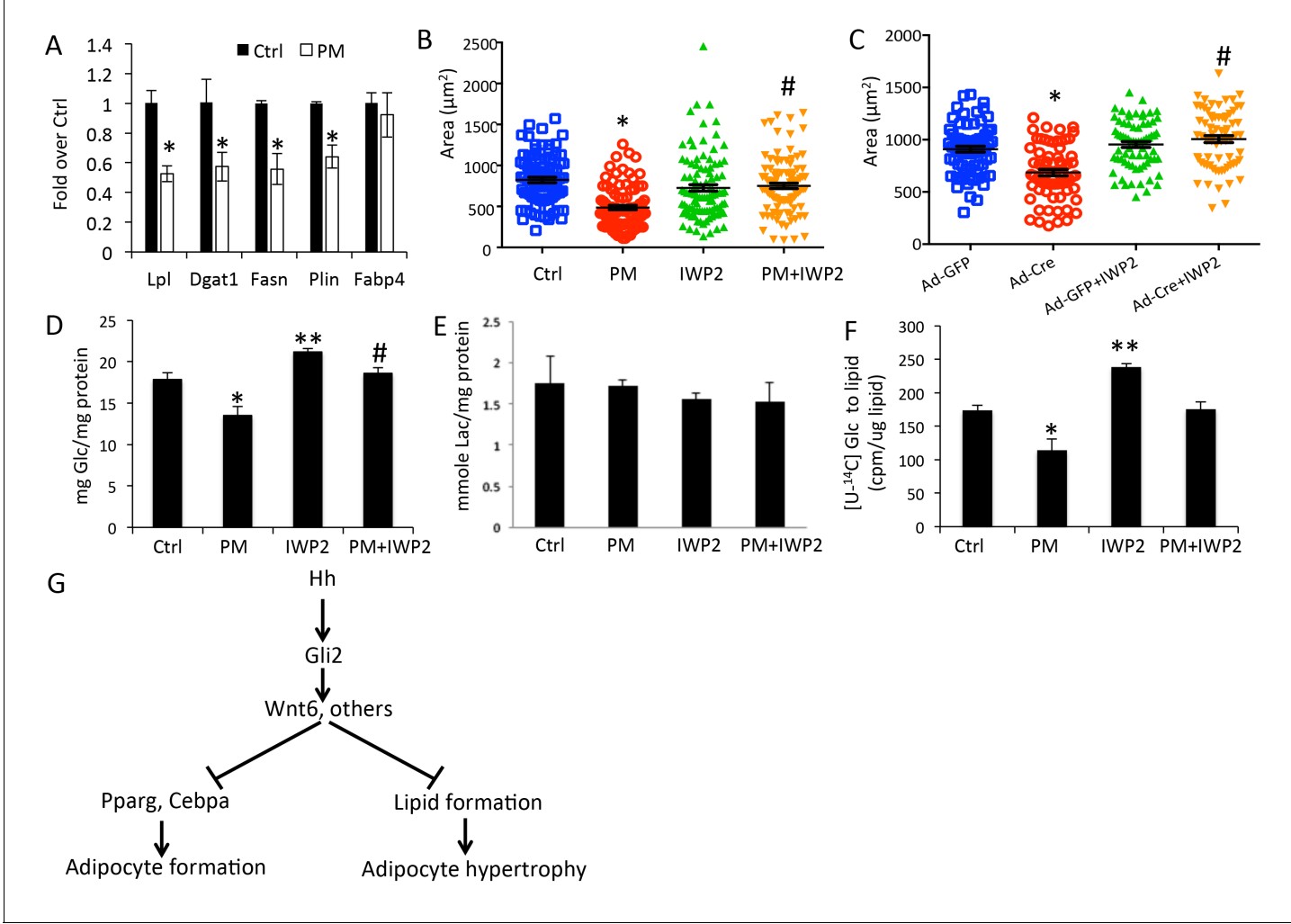

**Figure 7.** Hh signaling reduces glucose contribution to lipid in adipocytes. (**A**) PM suppressed lipogenesis genes but not adipocyte markers in adipocytes. (**B**) PM reduced but IWP2 rescued lipid droplet size in adipocytes. (**C**) IWP2 relieved the suppression of lipid droplet size by ΔNGli2 over-expression. (**D**) PM suppressed but IWP2 rescued glucose consumption by adipocytes. (**E**) No effect on lactate secretion by PM or IWP2. (**F**) PM suppressed but IWP2 partially rescued glucose contribution to lipid in adipocytes. *, **, #$p < 0.05$ for PM, IWP2 and PM-IWP2 interaction, respectively, based on two-way factorial ANOVA, n = 3. (**G**) A model depicting the role of Hh signaling in suppressing both adipogenesis and adipocyte hypertrophy.

DOI: https://doi.org/10.7554/eLife.31649.012

adipocyte differentiation but also lipid production by adipocytes. Molecularly, Gli2 is the principle transcription factor in the Gli family to mediate the anti-adipogenic and anti-lipogenic effects of Hh signaling. The study not only sheds new light on the mechanism of Hh signaling in adipogenesis but also provides a proof of principle that Hh activation may be explored for pharmaceutical treatment of obesity.

To our knowledge, the present study is the first to investigate the effect of Hh signaling specifi-cally in adult mice and in response to high fat diet. Although previous studies have identified an inhibitory role for Hh in adipose tissue formation during embryogenesis, it is necessary to assess its effect in adults and in response to dietary influences if the pathway were to be explored for thera-peutic purposes (*Pospisilik et al., 2010*; *Nosavanh et al., 2015*). Recent studies have demonstrated that both adipocyte de novo formation (hyperplasia) and hypertrophy contribute to fat accumulation in response to high fat diet, and that anatomically different fat depots employ the two mechanism differently (*Wang et al., 2013*). Because distinct adipose tissues differ substantially in their contribu-tion to whole-body nutrient metabolism, it is critical to evaluate whether Hh signaling differentially

affects the different fat depots. Here, we have found that Hh activation markedly suppressed adipocyte hypertrophy in both WAT and BAT in mice fed the high fat diet. Together with the studies in vitro, these results establish that Hh activation prevents obesity caused by high fat diet through inhibition of both adipocyte hyperplasia and hypertrophy.

The current study has provided new mechanistic insight about the inhibitory effect of Hh signaling on adipogenesis. The work distinguishes Gli2 from other members of the Gli family as the principale effector for Hh to suppress adipocyte differentiation. The downstream effectors for Gli2 in this process however, remain unresolved at present. Our cell culture work indicates that Wnt activation partly mediates the anti-adipogenic and anti-lipogenic functions of Hh signaling, but the physiological relevance of such a Hh-Wnt relay mechanism is yet to be tested in vivo. In this regard, it is worth noting that secreted frizzled-related protein 5, a secreted antagonist of Wnt proteins, was reported to stimulate adipocyte hypertrophy in obesity (*Mori et al., 2012*). In addition, previous studies have implicated Hes1 in mediating the anti-adipogenic function of Hh, and we have also observed the induction of Hes1 by PM in M2 cells (*Pospisilik et al., 2010*). Thus, it is likely that Hh-Gli2 signaling employs multiple effectors to suppress adipocyte formation and function.

Our results indicate that Hh may employ non-cell-autonomous mechanisms to suppress adipose tissue formation. Although Pparg-tTA targeted less than 50% of the adipocytes in either WAT or BAT, Hh activation in the targeted cells not only markedly diminished the overall dimension of all adipose depots, but also reduced the individual size of all adipocytes in WAT or BAT. Even more strikingly, Hh activation in less than 20% of bone marrow adipocytes essentially abolished all marrow fat. These results are consistent with the model that Hh activation in adipocyte-lineage cells induced expression of paracrine signals such as the Wnt proteins to suppress adipocyte differentiation and hypertrophy. Alternatively, Hh activation in a subset of the cells may lead to changes in circulating factors that in turn suppress fat accumulation systemically. Future experiments are necessary to distinguish those possibilities.

A major health burden of obesity is the dysregulation of whole-body glucose metabolism. Our result shows that by suppressing fat accumulation caused by HFD, Hh activation improved the systemic glucose metabolism in the mouse. Interestingly, even though Hh activation also suppressed adipose tissue accumulation in mice on regular chow diet, the lean mice did not exhibit any changes in glucose metabolism. This observation is similar to the previous study where Sufu was deleted with aP2-Cre, and indicates that Hh activation in the adipose tissues in a healthy state has minimal effects on general glucose metabolism. Thus, targeted activation of Hh signaling may be explored pharmaceutically to ameliorate the metabolic abnormalities associated with obesity.

## Materials and methods

### Mouse studies

The mouse strains of Pparg-tTA, TetO-Cre, R26-SmoM2 and R26-ΔNGli2 have been reported (*Joeng and Long, 2009*; *Kim et al., 2007*; *Jeong et al., 2004b*; *Perl et al., 2002*). The Animal Studies Committee at Washington University in St. Louis approved all mouse procedures. The experiments were performed with both male and female mice in a mixed genetic background of C57BL/6 (~70%) and 129 (~30%). Mice were fed normal chow (4% fat by weight, 11% calories from fat, Teklad) and doxycycline water (1 mg/ml) until two months of age and then exposed to high fat diet (21% fat by weight, 42% calories from fat, Harlan, cat# td.88137) with or without doxycycline water. Fat compositions were measured with EchoMRI. For GTT and ITT, D-glucose (1 g/kg mouse weight) or human insulin (0.75 unit/kg mouse weight) (Lilly, Indianapolis, Indiana), respectively, was injected intraperitoneally after 6 hr of starvation. Blood glucose levels were measured at the indicated intervals (*Li et al., 2000*). Each mouse was considered a biological replicate. All data points were included for analyses. For histology, adipose depots were fixed with formalin and embedded in paraffin before being sectioned at 6 μm thickness and stained with hematoxylin and eosin. For immunostaining of bone sections, femurs were fixed with 4% PFA overnight and decalcified for 3 days in 14% EDTA before being processed for sectioning in a cryostat machine. Perilipin antibody (#9349, Cell signaling) was used at 1: 100 dilution. Alexa fluor 594 goat anti-rabbit IgG (H + L) (#A11012, Invitrogen) secondary antibody was used at 1:200 dilution.

## Cell culture

M2-10B4 cells (cat# CRL-1972, ATCC) were maintained in RPMI-1640 (Gibco) with 10% fetal bovine serum (Gibco) as per ATCC instructions. The cell line was authenticated with the CO1 assay (interspecies) and tested free of mycoplasma contamination with Hoechst DNA stain (indirect) and Agar culture (direct) by ATCC. Unless otherwise indicated, M2 cells were seeded at the density of $1 \times 10^4$ cells/cm$^2$ for 48 hr before each experiment. Primary mouse embryonic fibroblasts were isolated from E13.5 embryos according to a published protocol (*Xu, 2005*; *Takahashi et al., 2007*). Adipogenic medium (AdipoM) contained 500 nM dexamethasone, 0.5 mM isobutylmethylxanthine, 50 µM indomethacin, and 1 µg/ml insulin (all from Sigma-Aldrich). For oil red O staining, after the cells were treated with AdipoM for 3 days, they were switched to the growth medium containing 1 µg/ml insulin (hereafter 'insulin-only media') for another 8 days with media change every 3–4 days. Ad-Cre and Ad-GFP were used at the 100 pfu/cell for infections (Viral Vector Core, University of Iowa). Oil red O staining was performed according to the described protocol (*Shi et al., 2012*). For knockdown experiments, shRNA lentivirus was used to infect cells for 24 hr before subsequent steps. The target sequences for Gli1 and Gli2 were described before (*Shi et al., 2015a*). The target sequence for Gli3 is as follows: 5'CCAATGAGAAACCGTATGTAT3'.

Preadipocytes were isolated as the stromal vascular fraction (SVF) from the gonadal fat pad according to a published protocol (*Hausman et al., 2008*). Adipogenic medium (AdipoM) for preadipocytes contained 500 nM dexamethasone, 0.5 mM isobutylmethylxanthine, 50 µM indomethacin, 1 µg/ml insulin and 10 µM troglitazone (all from Sigma-Aldrich). For qPCR analyses of differentiation markers, preadipocytes were incubated with AdipoM for 3 days. For oil red O staining and lipid droplet size measurements, preadipocytes were treated with AdipoM for 3 days and then incubated with the insulin-only media (lipid-accumulating phase) with or without PM or IWP2 for 11 days with media change every 3–4 days. For preadipocytes from R26$^{\Delta NGli2/+}$ mice, the cells were first induced for differentiation for 3 days and then incubated with the insulin-only medium containing either Ad-GFP or Ad-Cre and with or without the supplementation of IWP2 with media change every 3–4 days. The ImageJ software was used to measure the size of lipid droplets stained positive by oil red O. 100 droplets were measured for each treatment in each experiment and three independent experiments were conducted. Representative results from one experiment were presented here. IWP2 (Sigma) was used at 5 µM dissolved in DMSO. PM (540223, Calbiochem) was used at 1 µM dissolved in DMSO.

## RT-qPCR

Total RNA was isolated with QIAGEN RNeasy kit (#74104, QIAGEN) and transcribed into cDNA with iScript cDNA synthesis kit (Bio-Rad). Fast-start SYBR Green (Bio-rad) was used for qPCR in Step-One machine (ABI). Nucleotide sequence of primers is listed in *Table 1*. 18S rRNA was used for normalization. Each RNA sample extracted from one cell culture plate (well) or one mouse was considered a biological replicate.

## Lipid extraction and metabolic measurements

For testing glucose incorporation to lipids, preadipocytes isolated from 8-week-old C57BL/6J mice were treated with AdipoM for 3 days before being switched to insulin-only medium with or without PM or IWP2 for additional 8 days. Then, the uniformly labeled [U-$^{14}$C$_6$] glucose (cat# ARC 0122G, American Radiolabeled Chemicals, St. Louis, MO, USA) was added at 0.2 µl/ml to the medium, with or without PM or IWP2, for three more days. Lipids were extracted with Lipid Extraction Kit (STA-612, Cell biolabs), and an equal volume of the extract was measured either with Lipid Quantification Kit (STA-613, Cell biolabs) or for radioactivity with a scintillation counter. The amount of $^{14}$C radioactivity was normalized to lipid quantity (cpm/µg). Glucose and lactate measurements were done as described (*Esen et al., 2013*). Briefly, for glucose consumption measurements, preadipocytes were incubated with AdipoM for 3 days and then switched to insulin-only medium with or without PM or IWP2 for three more days. Aliquots of the media and glucose standards were assayed with Glucose (HK) Assay Kit (Sigma catalog number GAHK20) and read at 340 OD using a plate reader (BioTek model SAMLFTA, Gen5 software). Lactate was measured with L-lactate assay kit from Eton biosciences (catalog number 1200011002). Each cell culture well was considered a biological replicate.

**Table 1.** Nucleotide sequence of primers.

| pene | Primer F/R | Sequence 5′ to 3′ |
| --- | --- | --- |
| 18S | F | CGGCTACCACATCCAAGGAA |
| 18S | R | GCTGGAATTACCGCGGCT |
| Glil | F | TACCATGAGCCCTTCTTTAGGA |
| Glil | R | GCATCATTGAACCCCGAGTAG |
| Gli2 | F | CACCTGCATGCTAGAGGCAAA |
| Gli2 | R | AGAAGTCTCCATCTCAGAGGCTCATA |
| Gli3 | F | CCCTGCATTGAGCTTCACCTA |
| Gli3 | R | AATGCGGAGCCTAAGCTTTG |
| Ptchl | F | GCCTTGGCTGTGGGATTAAAG |
| Ptchl | R | CTTCTCCTATCTTCTGACGGGT |
| c/ebp alpha | F | GAATCTCCTAGTCCTGGCTC |
| c/ebp alpha | R | GATGAGAACAGCAACGAGTAC |
| c/ebp beta | F | GCCACGGACACCTTCGAGG |
| c/ebp beta | R | CGGCTCCGCCTTGAGCTG |
| c/ebp delta | F | CGACTTCAGCGCCTACATTGA |
| c/ebp delta | R | CTAGCGACAGACCCCACA |
| Ppar gamma | F | GGAAAGACAACGGACAAATCAC |
| Ppar gamma | R | TACGGATCGAAACTGGCAC |
| Fabp4 | F | CGGCCCAATCCTATCCTGGA |
| Fabp4 | R | AGGTTGAAGTGGGTCAAGCAA |
| Wnt5a | F | GCGTGGCTATGACCAGTTTAAGA |
| Wnt5a | R | TTGACATAGCAGCACCAGTGAA |
| Wnt6 | F | GGTTTACACCAGCCCACGAA |
| Wnt6 | R | GCAACTAGCAAAGGGCCTTTC |
| Wnt9a | F | CGTGGGTGTGAAGGTGATAAG |
| Wnt9a | R | GCAGGAGCCAGACACACCAT |
| Nkd2 | F | CTTTCTGGGACGACAAGGGTT |
| Nkd2 | R | AGTGCGTCAATGTTCAAGTGC |
| Ucp1 | F | AGGCTTCCAGTACCATTAGGT |
| Ucp1 | R | CTGAGTGAGGCAAAGCTGATTT |
| Cidea | F | TGACATTCATGGGATTGCAGAC |
| Cidea | R | GGCCAGTTGTGATGACTAAGAC |
| Lpl | F | GGGAGTTTGGCTCCAGAGTTT |
| Lpl | R | TGTGTCTTCAGGGGTCCTTAG |
| Fasn | F | AAGTTGCCCGAGTCAGAGAA |
| Fasn | R | CGTCGAACTTGGAGAGATCC |
| Plin | F | CTGTGTGCAATGCCTATGAGA |
| Plin | R | CTGGAGGGTATTGAAGAGCCG |
| Dgat1 | F | GTGTGTGGTGATGCTGATCC |
| Dgat1 | R | GATGCAATAATCACGCATGG |
| Axin2 | F | TGAGCGGCAGAGCAAGTCCAA |
| Axin2 | R | GGCAGACTCCAATGGGTAGCT |

DOI: https://doi.org/10.7554/eLife.31649.013

## Statistical and power analyses

All quantitative data are presented as mean ±SD with a minimum of three independent samples. Statistical significance is determined by Student's t test or two-way factorial ANOVA (http://vassarstats.net/). The minimal sample size of 3 was calculated according to http://www.stat.ubc.ca/~rollin/stats/ssize/n2.html, with 25% difference in mean values, 10% standard deviation and the default $\alpha$ (0.05) and power (0.80) values.

## Acknowledgements

The work is supported by R01 grants DK111212 and AR060456 (FL). EchoMRI experiments were performed at Washington University Diabetes Research Center, supported by NIH (P30 DK020579). We thank Dr. Irfan Lodhi for advice on preadipocyte cultures.

## Additional information

### Funding

| Funder | Grant reference number | Author |
|---|---|---|
| National Institute of Diabetes and Digestive and Kidney Diseases | DK111212 | Fanxin Long |
| National Institute of Arthritis and Musculoskeletal and Skin Diseases | AR060456 | Fanxin Long |

The funders had no role in study design, data collection and interpretation, or the decision to submit the work for publication.

### Author contributions

Yu Shi, Data curation, Investigation, Methodology, Writing—original draft; Fanxin Long, Conceptualization, Formal analysis, Supervision, Funding acquisition, Writing—review and editing

### Author ORCIDs

Fanxin Long ⓘ https://orcid.org/0000-0001-9785-5379

### Ethics

Animal experimentation: This study was performed in strict accordance with the recommendations in the Guide for the Care and Use of Laboratory Animals of the National Institutes of Health. All of the animals were handled according to approved institutional animal care and use committee (IACUC) protocol (#20170126) of Washington University in St. Louis.

### Decision letter and Author response

Decision letter https://doi.org/10.7554/eLife.31649.017
Author response https://doi.org/10.7554/eLife.31649.018

## Additional files

### Supplementary files

• Supplementary file 1. RNA-seq data for Hh signaling in M2-10B4 cells showing genes with a minimum of 2 fold change in mRNA level in response to PM after 72 hr.
DOI: https://doi.org/10.7554/eLife.31649.014

• Transparent reporting form
DOI: https://doi.org/10.7554/eLife.31649.015

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
