## [Decision Letter]

[Editors’ note: a previous version of this study was rejected after peer review, but the authors submitted for reconsideration. The first decision letter after peer review is shown below.]

Thank you for submitting your work entitled "Hedgehog signaling via *Gli2* prevents obesity induced by high fat diet in adult mice" for consideration by *eLife*. Your article has been reviewed by three peer reviewers, and the evaluation has been overseen by a Reviewing Editor and a Senior Editor. The reviewers have opted to remain anonymous.

Our decision has been reached after consultation between the reviewers. Based on these discussions and the individual reviews below, we regret to inform you that your work as it stands will not be considered further for publication in *eLife*. It is the consensus opinion of the reviewers that this study nicely builds upon previous work in cultured cells and mice which established that activation of Hh signaling leads to inhibition of adipogenesis in both white and brown adipose tissue. The novel contribution of this study is that Hh signaling can protect mice from obesity and its related metabolic complications. In vivo and in vitro data suggest that specific activation of Hh signaling in adipose tissue of postnatal mice have a direct effect on diet-induced obesity. This activation also showed significant improvement of glucose tolerance and insulin sensitivity implying that adipose tissue specific activation of Hh can affect glucose utilization (uptake, transport, or consumption) in tissues other than adipose tissue. This inhibitory effect appears to be mediated by the Hh downstream effector, *Gli2*, likely through activation of the *Gli2* target gene, Wnt6. These are significant findings of general interest.

However, despite the interesting set of experiments, the paper has several weaknesses relating to the Dox treatments and alleles used, level of modification of Wnt singaling, and presentation which we anticipate will require extensive reanalysis and the generation of additional data. Since it is *eLife* policy to only invite revisions in cases where revisions can be reasonably completed within two months, we cannot invite revision of the current manuscript. However, we are in principle interested in the study and we would be enthusiastic about reviewing a substantially revised version that addresses our concerns, with the proviso that it will be treated as a new submission. Such a submission would not necessarily be seen by the same reviewers and thus additional concerns might be raised. *eLife* is highly selective, which means that the majority of submissions are rejected, but we thank you for sending your work for review and we hope you will submit to *eLife* again in the future.

Reviewer #1:

In this manuscript, Shi et al. use a conditional approach to activate Hh signaling in a subset of adult WAT and BAT cells or their precursors (Figure 1). Hh pathway activation has a mild effect on reducing adiposity (Figure 2, Figure 7). In vitro, Hh pathway activation inhibits adipogenesis, as has previously been described (Figure 5). Knockdown studies suggest that this in vitro inhibition is mediated through *Gli2*, the principal transcriptional activator used by Hh signaling (Figure 6). *Gli2* is able to induce Wnt6 (Figure 8) and inhibition of Wnt secretion modestly reverses the *Gli2*-mediated inhibition of adipogenic gene expression in vitro (Figure 9E). In addition to affecting adipogenesis, Hh pathway activation in vitro has mild effects on fat formation in those adipocytes that do form (Figure 10).

Several previous studies have demonstrated that the Hh pathway is a strong anti-adipogenic pathway including two in vivo studies, in which adipose-specific Hh activation completely inhibits WAT (Pospisilik, 2010) and BAT (Nosavanh, 2015) formation. This study complements these previous data nicely by suggesting that Hh activation can also inhibit diet-induced obesity in adult mice. This group then carefully analyzed the downstream cascade and convincingly argue for *Gli2* as the main effector of pathway activation in the fat.

Major concerns:

The Ppary-tTA allele used here has been shown to cause lipodystrophy in the absence of Dox (Kim, 2007, PNAS). Moreover, Dox treatment suppresses these pathologies, suggesting that the observed lipodystrophy results from promiscuous transcriptional activity of tTA. Since the phenotype reported in this manuscript is that fat mass is reduced in the absence of Dox, an alternative explanation could be that as soon as Dox is removed, the anti-adipogenic effects of this allele are activated, thereby counteracting the fat mass gain due to the high fat diet. To separate their findings from the anti-adipogenic role of the Ppary-tTA line, the authors should repeat the experiment with mice just carrying the Ppary-tTA allele {plus minus} Dox.

Similarly, many different classes of antibacterial agents, including tetracyclines, promote animal growth. Subtherapeutic levels of chlortetracycline causes increased adiposity and metabolic alterations in mice (Cho et al.,). Since the phenotype reported in this manuscript is that fat mass is reduced in the absence of Dox, an alternative explanation could be that removal of Dox removes this recognized pro-adipogenic influence. To separate their findings from the pro-adipogenic role of Dox, the authors should repeat the experiment with mice lacking one of the three transgenic alleles {plus minus} Dox.

Given that Wnt6 is induced by Hh signaling 17-fold in M2 cells, 1500-fold in MEFs but only 3-fold in vivo (Figure 8E), and IWP2 has a modest effect (less than twofold on Pparg expression) on preadipocytes, and certainly doesn't restore the robust inhibition of adipogenesis caused by purmorphamine (Figure 9D vs E), it isn't clear that Wnt signaling is the primary means by which Hh signaling inhibits adipogenesis. Some confirmation of the effect on Wnt signaling in vivo by Western of animal tissue and analysis of Wnt signaling in fat pads would lend important support to this crucial point. Similarly, is Wnt6 induced in the SmoM2 model, as one would expect?

*Reviewer #2:*

That activation of Hh signaling inhibits adipogenesis (both in white and brown adipose tissue) is well established in cultured cells and in mice. However, whether this can protect mice from obesity and its related metabolic complication has not yet been reported. In this paper, Shi and Long presented a comprehensive set of in vivo and in vitro data to show that adipose tissue specific activation of Hh signaling in postnatal mice not only reduces adiposity but also improves glucose tolerance and insulin sensitivity in diet-induced obese mice. This inhibitory effect is mainly carried out by the Hh downstream effector, *Gli2*. These authors further showed that both differentiation and lipid accumulation were inhibited by Hh signaling, likely through activation of a *Gli2* target gene, Wnt6. Wnt6 expression is dependent on Hh signaling in M2 cells, MEFs, and adipose tissue. Blocking palmitoylation of Wnt6 reduced Wnt signaling activity and alleviated the inhibitory effect of Hh signaling on adipogenesis.

As far as I know, this is the first paper to provide genetic evidence that activation of Hh signaling in mice can have a direct effect on diet-induced obesity. Because Hh signaling can inhibit adipogenesis in cultured cells, it's no surprise to see that mutant mice with high Hh pathway activity in adipose tissue are leaner even under high fat-diet treatment. However, the fact that these mice also showed significant improvement of glucose tolerance and insulin sensitivity is impressive and suggests that adipose tissue specific activation of Hh can affect glucose utilization (uptake, transport, or consumption) in tissues other than adipose tissue. In my view, this is the most important finding from this study and needs to be further characterized in this paper. For example, gene expression analysis of skeletal muscle, liver, or subcutaneous adipose tissue would provide insight into the expression of genes responsible for glucose uptake and insulin pathway activation in these tissues. Therefore, it is somewhat disappointing that the authors focused on characterizing the effect of Hh on inhibiting adipogenesis, which has been previously reported by several studies rather than on how Hh regulates glucose tolerance and insulin insensitivity through controlling adipose tissue formation in high-fat diet fed mice.

Overall, the data presented in this paper is well organized and easy to understand, and the genetic data is convincing and nicely shows the epistatic relationship between Smo and *Gli2*. However, the in vitro differentiation assays appear not to be working very well. In all the cell lines (M2, MEFs, and primary preadipocytes) examined, adipogenic induction (control cells) only resulted in the differentiation of a low% of cells, making it difficult to draw definitive conclusions about the extent to which differentiation is disrupted because most of the cells did not respond to the adipogenic induction. Perhaps adjusting the concentration of the adipogenic induction media components will in the first place improve the efficiency of differentiation and provide more definitive results to support the author's conclusions.

Reviewer #3:

In this study, Shi and Long report that activation of Hedgehog pathway reduced the development of obesity among mice fed a high fat diet. This repression occurs through *Gli2*, at the transcriptional level. In ChIP-seq experiments in a cell culture system, they find that *Gli2* binds to several Wnt loci, and knockdown of *Gli2* prevents Hedgehog stimulated activation of these Wnts. The pharmacological inhibition of Wnt proteins largely (but not completely) prevents Hedgehog-mediated reduction of adipogenesis in cultured cells, suggesting that Wnts are the major mediator of *Gli2*-mediated suppression of adipogenesis. This model is consistent with their earlier data suggesting that Hh-repression occurs cell autonomously.

This study is interesting and logically presented but I do have one major concern. The connection between Hedgehog signaling and Wnt activation, a major conclusion of the study, seems somewhat contrived, relying on 'cherry-picked' ChIP binding data that (see below) and cell culture experiments with different Gli and Wnt kinetics that are hard to compare with the in vivo models. Clarifying the transcriptional relationship between Hedgehog and Wnts in these systems is critical for solidifying this connection.

1) As I understand it, the current data suggest that Hedgehog inhibits adipogenesis but whether it would cause an already obese mouse to lose weight remains unaddressed. While I realize that doing this experiment is beyond the scope of the study, I raise this concern because the authors are making the claim that Hedgehog pathway activation could potentially be used to treat obesity. If my understanding is correct, the authors should consider toning-down / or qualifying statements like "Thus, targeted activation of the Hh pathway may be explored to combat diet-induced obesity."

2) Why is the magnitude of Hh-mediated response so different between high fat and regular diets for SmoM2? Both Gli1 and Ptch1 have higher responses in the RosaSmoM2 and RosaGli2deltaN dominant active genetic systems compared to the SmoM2 in mice maintained on a normal diet. Is this a confounding biological factor?

3) Related to point #2, it is not clear in Figure 9D how long pre-adipocytes were cultured without DOX before assaying by qRT-PCR. There, the difference in Gli1 induction (4-fold) is really striking compared to the in vivo Gli1 response using RosaGli2deltaN system, which appears at least 25-fold higher (c.f. Figure 7). This makes me concerned about whether this assay really effectively models the Hh-mediated repression observed in 16 week of the high fat diet regimen. If it does not, then the subsequent Wnt inhibitor assays used to show a loss of Hedgehog-mediated suppression of adiopogenesis markers might also be hard to compare meaningfully in vivo.

4) The fat pads from the activated *Gli2*-deltaN activated mice have no upregulation Wnt 5a or Wnt 9a. While they do have significant upregulation of Wnt 6 (3-fold; Figure 8E) it is much lower than the values obtained in the cell culture assays. Would this level of Wnt really be sufficient to account for cell autonomous Gli-mediated repression? Is this discrepancy due to negative feedback from prolonged Wnt signaling, that the in vitro systems are not effectively modeling the in vivo response, or that Wnts are, despite the ChIP data, not directly regulated by *Gli2* in the in vivo systems?

5) The *Gli2* ChIP data provided for review is incomplete and does not meet current standards for providing all raw data. The ChIP binding coordinates are not described in the excel file that is provided, the raw data (ChIP-seq reads) must be deposited in GEO with an accession number and reviewer access provided for reviewer / editorial access. In addition, the currently provided dataset (which should ideally be deposited both in GEO as a Supplementary file) should list the ChIP coordinates. The methods for analyzing the ChIP data need to be clarified. For example, what does the P-value in the table refer to? Are these peaks enriched relative to an input or to a cell line not expressing Flag? Are enhancers for known Hedgehog pathway targets (Ptch1, Gli1) also bound by *Gli2*?

6) Besides the known roles for Wnts in inhibiting adipogenesis, it is not clear why the focus is on the three Wnt genes for the ChIP experiment. In my quick examination of the predicted target genes provided in the excel spreadsheet using an online KEGG enrichment analysis tool, the most upregulated pathways were (in order of being most statistically enriched: Jak-Stat (25 genes), Notch (13 genes), Tgf-β (16 genes), Wnt (19 genes). This cursory analysis suggests that a strong rationale for focusing exclusively on Wnt needs to be provided. A confounding factor in the current level of data analysis is that, if my understanding is correct, the nearest gene in the genome is shown on the dataset regardless of whether it is transcriptionally active in adipocytes. Could the ChIP regions be intersected with Hedgehog-responsive genes in adipocytes (or if this is unavailable at least genes known to be transcribed in adipocytes) to predict which of the ChIP regions are functional?

7) Related to concern 4, I am concerned that these Wnt sites may not represent direct Gli targets. The ability the Gli-bound sites to act as enhancers should be tested in cell culture enhancer assays in the presence or absence of Hedgehog pathway activation.

8) Please indicate the genetic background of the mice in the experiments.

9) Are there sex-specific differences in Hh-mediated inhibition of obesity? If so, they should be shown. If not, this should be stated.

10) Figure 9A-C: What is the time course of the Wnt inhibition experiments? Please provide appropriate details in the Materials and methods / Results sections.

[Editors’ note: what now follows is the decision letter after the authors submitted for further consideration.]

Thank you for resubmitting your work entitled "Hedgehog signaling via *Gli2* prevents obesity induced by high fat diet in adult mice" for further consideration at *eLife*. Your revised article has been evaluated by Fiona Watt (Senior editor) and two reviewers, one of whom evaluated the previous version.

Summary:

The study by Long and Shi offers novel insights into a role of Hh signalling in adult adipose tissue physiology and diet-induced obesity. Although a mouse model used in the study provides limited efficiency in targeting different fat depots, it shows strong anti-adipogenic and anti-lipogenic effects of Hh signalling activation in high-fat diet condition. The lineage tracing experiments using Pparg-tTA;TetO-Cre in combinations with R26-SmoM2 and R26-ΔNGli2/+, separately, are both convincing of a SHH mediated effect on adult adipogenesis, high-fat diet induced obesity and suppressing metabolic dysfunctions as assessed by glucose tolerance and insulin sensitivity. The in vitro data using primary adipocytes is also convincing of SHH-mediating Wnt induction. RNA from whole gonadal fat pad from both transgenic lines overexpressing SHH or *Gli2* shows Wnt6 induction. Primary preadipocytes isolated from gonadal fat pad of PPAR ΔNGli2/+ and in the presence of doxycycline and IWP2 induce Pparg and Fabp4 adipocyte markers. This is an interesting study that the authors have strengthened since the original submission.

Essential revisions:

The major concern remains the link between Hedgehog signalling and Wnt activation, since the results do not strongly support the authors' conclusions. These concerns are expanded in the specific comments below. While the data strongly suggest that *Gli2* regulates Wnt activity, they do not provide compelling support the direct transcriptional regulation of Wnt proteins by *Gli2*.

1) The enhancer experiments shown in Figure 5 are not convincing. First, the two sites are less than 2-fold activated. For in vitro GLI enhancer assays, this is not a compelling increase in activity. It is possible that this is because the M2 cells were only treated with an unspecified amount of purmorphamine for 24 hours when previous work by this lab showed maximal stimulation was achieved at 48 hours (Shi et al., 2015). Alternatively, perhaps the M2 bone marrow line does not model adipogenic responses under the current culture conditions, or that Wnt signaling is not active under these biological conditions. Finally, it is certainly still possible that these regions are not biological enhancers. Taken together, these experiments do not adequately support their conclusion that Wnt6 as a direct transcriptional target of *Gli2*. I can think of other approaches that could be used to test this (in vivo transgenic enhancer assays or possibly CRISPR-based deletions that are clearly outside the scope of the current work and might ultimately end up showing no enhancer activity.

2) The ChIP-seq data in Figure 5 shows relatively weak enrichment of *Gli2* at the Wnt6 region compared to binding at Gli1 or Wnt9a (Figure S5). This, combined with the fact that the ChIP data was acquired from a lentivirally driven *Gli2* construct in M2 cells does not provide high levels of confidence that these represent *Gli2* binding in adipocytes.

3) Discussion section "Furthermore, *Gli2* functions at least partly through direct transcriptional regulation of several Wnt proteins"; Discussion section "Further downstream, several Wnt genes are direct transcriptional targets of *Gli2*." Both of these statements convey the erroneous perception that Gli22 directly regulates multiple Wnt proteins. Their own data suggests that Wnt5a and 9b may not be in vivo targets (as they themselves acknowledge in subsection “Wnt6 is a direct target of Hh-*Gli2* signaling”). Thus, at best, it could be claimed that *Gli2* regulates the transcriptional regulation of a Wnt protein. However, the evidence that *Gli2* regulates Wnt6 is unconvincing.

4) The new supplemental spreadsheet showing ChIP-seq reads is helpful. However, both within the supplemental dataset and subsection “ChIP-seq experiments”, there is no statistical information on the statistical analysis done for peak calling (stated that this is done in Partek but this is not sufficient information for a reviewer or reader to know the threshold metric used for calling a peak). Similarly, in the supplemental spreadsheet, there is a 'scaled fold change'. While it is not mentioned how this was calculated, it seems like it is a modified read count in dox treated versus dox untreated. The authors should calculate a modified P-value to reflect the quality of these peaks.

5) The authors need to tone down their claims of a direct transcriptional link between Shh and Wnt, and remove the data that the reviewers consider to be weak (see above).

[Editors' note: further revisions were requested prior to acceptance, as described below.]

Thank you for resubmitting your work entitled "Hedgehog signaling via *Gli2* prevents obesity induced by high fat diet in adult mice" for further consideration at *eLife*. Your revised article has been evaluated by Fiona Watt (Senior editor) and the two original reviewers.

The manuscript has been improved but there are some remaining issues that need to be addressed before acceptance, as outlined below. Specifically, the reviewers request that you remove all the ChIP-seq data from the main body of the paper and the abstract. In addition, since none of the in vitro assays appears to be adipogenic, the manuscript lacks a solid connection between the in vivo findings and the downstream molecular mechanism studies and you should address this within the text.

As highlighted previously there are several problems with your ChIP-seq data: 1) there are many binding regions that do not have enhancer activity – so this binding does not necessarily imply that the regulation is direct, and 2) the ChIP-seq results were obtained in M2 cells.

---

## [Author Response]

[Editors’ note: the author responses to the first round of peer review follow.][…] Reviewer #1:[…] The Ppary-tTA allele used here has been shown to cause lipodystrophy in the absence of Dox (Kim, 2007, PNAS). Moreover, Dox treatment suppresses these pathologies, suggesting that the observed lipodystrophy results from promiscuous transcriptional activity of tTA. Since the phenotype reported in this manuscript is that fat mass is reduced in the absence of Dox, an alternative explanation could be that as soon as Dox is removed, the anti-adipogenic effects of this allele are activated, thereby counteracting the fat mass gain due to the high fat diet. To separate their findings from the anti-adipogenic role of the Ppary-tTA line, the authors should repeat the experiment with mice just carrying the Ppary-tTA allele {plus minus} Dox.

We appreciate the reviewer’s concern and have now repeated the experiment with mice just carrying the Pparg-tTA allele as recommended. Specifically, the Pparg-tTA mice were raised with Dox from conception to 2 months of age, and then fed HFD with or without Dox water for 2 additional months. We have found no Dox effect on body composition, body weight or GTT. Please see the data in the supplemental figures (Figure S1C-E).

Similarly, many different classes of antibacterial agents, including tetracyclines, promote animal growth. Subtherapeutic levels of chlortetracycline causes increased adiposity and metabolic alterations in mice (Cho et al.,). Since the phenotype reported in this manuscript is that fat mass is reduced in the absence of Dox, an alternative explanation could be that removal of Dox removes this recognized pro-adipogenic influence. To separate their findings from the pro-adipogenic role of Dox, the authors should repeat the experiment with mice lacking one of the three transgenic alleles {plus minus} Dox.

We have now repeated the experiment as suggested. Specifically, wild type mice without any of the transgenic alleles were raised with Dox from conception to 2 months of age, and then fed HFD with or without Dox water for 2 additional months. We detected no Doc effect on body weight or body composition (Figure S1A, B).

Given that Wnt6 is induced by Hh signaling 17-fold in M2 cells, 1500-fold in MEFs but only 3-fold in vivo (Figure 8E), and IWP2 has a modest effect (less than twofold on Pparg expression) on preadipocytes, and certainly doesn't restore the robust inhibition of adipogenesis caused by purmorphamine (Figure 9D vs E), it isn't clear that Wnt signaling is the primary means by which Hh signaling inhibits adipogenesis. Some confirmation of the effect on Wnt signaling in vivo by Western of animal tissue and analysis of Wnt signaling in fat pads would lend important support to this crucial point. Similarly, is Wnt6 induced in the SmoM2 model, as one would expect?

We agree that Hh signaling likely engages multiple downstream effectors besides Wnt6 to inhibit adipogenesis in vivo (see Discussion section). As per the reviewer’s suggestion, we have now performed additional experiments to confirm the involvement of Wnt signaling. Specifically, we show that a common Wnt target gene Axin2 was up-regulated in the gonadal fat pad upon ΔNGli2 expression (Figure 5). We further show that in the SmoM2 model Wnt6 was also upregulated by ~4 fold in the gonadal fat pad (Figure 5).

Reviewer #2:As far as I know, this is the first paper to provide genetic evidence that activation of Hh signaling in mice can have a direct effect on diet-induced obesity. Because Hh signaling can inhibit adipogenesis in cultured cells, it's no surprise to see that mutant mice with high Hh pathway activity in adipose tissue are leaner even under high fat-diet treatment. However, the fact that these mice also showed significant improvement of glucose tolerance and insulin sensitivity is impressive and suggests that adipose tissue specific activation of Hh can affect glucose utilization (uptake, transport, or consumption) in tissues other than adipose tissue. In my view, this is the most important finding from this study and needs to be further characterized in this paper. For example, gene expression analysis of skeletal muscle, liver, or subcutaneous adipose tissue would provide insight into the expression of genes responsible for glucose uptake and insulin pathway activation in these tissues. Therefore, it is somewhat disappointing that the authors focused on characterizing the effect of Hh on inhibiting adipogenesis, which has been previously reported by several studies rather than on how Hh regulates glucose tolerance and insulin insensitivity through controlling adipose tissue formation in high-fat diet fed mice.

We appreciate that the reviewer recognizes the significance of our finding. As the anti-obesity effect of Hh activation in the face of high fat diet is of potential translational value, we have focused our effort on gaining more mechanistic insights about the effect. The studies have uncovered an Hh-*Gli2*-Wnt6 signaling axis as an important part of the anti-adipogenic mechanism.

Besides the reduced adiposity, Hh activation also achieved impressive metabolic benefits in insulin signaling and glucose handling. We believe that the metabolic benefits are secondary to the reduced obesity but not necessarily specific to Hh signaling. We agree with the reviewer that a mechanistic understanding of the link between adiposity and whole-body metabolism is critical, but believe that such a fundamental question warrants a separate study.

Overall, the data presented in this paper is well organized and easy to understand, and the genetic data is convincing and nicely shows the epistatic relationship between Smo and Gli2. However, the in vitro differentiation assays appear not to be working very well. In all the cell lines (M2, MEFs, and primary preadipocytes) examined, adipogenic induction (control cells) only resulted in the differentiation of a low% of cells, making it difficult to draw definitive conclusions about the extent to which differentiation is disrupted because most of the cells did not respond to the adipogenic induction. Perhaps adjusting the concentration of the adipogenic induction media components will in the first place improve the efficiency of differentiation and provide more definitive results to support the author's conclusions.

We appreciate the reviewer’s concern. We think the relatively insufficient adipogenesis (judged by oil red O staining) of M2 cells may reflect their nature as bipotent mesenchymal progenitors for osteoblast and adipocytes. In addition, the adipogenic media that we used for M2 and MEFs did not contain troglitazone that could have produced more robust adipogensis. The primary preadipocytes however did exhibit robust differentiation in our assays, although we did not include images of oil red O staining in the paper for the sake of brevity. Regardless, in all cases in addition to quantifying the number of oil red O-stained cells, we have performed qPCR for several molecular markers to assess adipogenesis quantitatively (Figure 3, Figure 6). Therefore, we feel confident about the conclusions drawn from those experiments.

Reviewer #3:[…] 1) As I understand it, the current data suggest that Hedgehog inhibits adipogenesis but whether it would cause an already obese mouse to lose weight remains unaddressed. While I realize that doing this experiment is beyond the scope of the study, I raise this concern because the authors are making the claim that Hedgehog pathway activation could potentially be used to treat obesity. If my understanding is correct, the authors should consider toning-down / or qualifying statements like "Thus, targeted activation of the Hh pathway may be explored to combat diet-induced obesity."

We appreciate the reviewer’s concern and agree that future experiments are necessary to determine whether Hh activation is sufficient to reverse diet-induced obesity. We have deleted the statement in question.

2)Why is the magnitude of Hh-mediated response so different between high fat and regular diets for SmoM2? Both Gli1 and Ptch1 have higher responses in the RosaSmoM2 and RosaGli2deltaN dominant active genetic systems compared to the SmoM2 in mice maintained on a normal diet. Is this a confounding biological factor?

We thank the reviewer for the astute observation. We do not know for certain why this is the case at the moment. As targeting of the adipocytes by our strategy was < 50% in mice on the regular diet (Figure 1), it is possible that the high fat diet caused more adipocytes to be targeted by PpargtTA;tetoCre, resulting in the apparent increase in Hh response when the fat pad was assayed as a whole. It is also possible that the high fat diet somehow increased the transcriptional output of Hh signaling, which in itself would be very interesting to investigate in the future. Although the difference in question could help to explain why the mutant mice on the regular diet did not show a phenotype until much later (26 weeks) than those on high fat diet (8 weeks) after Dox withdrawal, it should not affect our main conclusion that Hh activation reduces adiposity in response to high fat diet.

3) Related to point #2, it is not clear in Figure 9D how long pre-adipocytes were cultured without DOX before assaying by qRT-PCR. There, the difference in Gli1 induction (4-fold) is really striking compared to the in vivo Gli1 response using RosaGli2deltaN system, which appears at least 25-fold higher (c.f. Figure 7). This makes me concerned about whether this assay really effectively models the Hh-mediated repression observed in 16 week of the high fat diet regimen. If it does not, then the subsequent Wnt inhibitor assays used to show a loss of Hedgehog-mediated suppression of adiopogenesis markers might also be hard to compare meaningfully in vivo.

The preadipocytes were cultured with or without Dox for three days before assaying by qRTPCR. The relatively low induction of Gli1 compared to that in vivo could be due to the short duration of Dox withdrawal, or the small percentage of cells expressing Pparg-tTA due to cell heterogeneity in the preadipocyte preparation (i.e., stromal vascular fraction of the adipose tissue). It is also possible that the transcriptional response to *Gli2* in preadipocytes is less robust than that in the more mature adipocytes that are only present in the intact adipose depot. We acknowledge the limitations of the in vitro system, and that the in vivo anti-adipogenic effect of Hh signaling may be mediated by additional mechanisms besides Wnt induction.

4) The fat pads from the activated Gli2-deltaN activated mice have no upregulation Wnt 5a or Wnt 9a. While they do have significant upregulation of Wnt 6 (3-fold; Figure 8E) it is much lower than the values obtained in the cell culture assays. Would this level of Wnt really be sufficient to account for cell autonomous Gli-mediated repression? Is this discrepancy due to negative feedback from prolonged Wnt signaling, that the in vitro systems are not effectively modeling the in vivo response, or that Wnts are, despite the ChIP data, not directly regulated by Gli2 in the in vivo systems?

These are all very thoughtful questions. The reviewer raises an interesting negative feedback mechanism that could potentially explain the muted response of Wnt expression in vivo. It is also possible that Wnt 6 is induced by *Gli2* only in the preadipocytes that may be a small constituency of the adipose depot in vivo. As per the reviewer’s suggestion, we have now performed in vitro luciferase reporter assays to assess the Gli-bound sites in mediating the transcriptional activation by Hh signaling. The results show that two out of the four sites exhibit measurable activities (Figure 5). We therefore conclude that Wnt6 is likely a direct transcriptional target for *Gli2* in preadipocytes.

5) The Gli2 ChIP data provided for review is incomplete and does not meet current standards for providing all raw data. The ChIP binding coordinates are not described in the excel file that is provided, the raw data (ChIP-seq reads) must be deposited in GEO with an accession number and reviewer access provided for reviewer / editorial access. In addition, the currently provided dataset (which should ideally be deposited both in GEO as a Supplementary file) should list the ChIP coordinates. The methods for analyzing the ChIP data need to be clarified. For example, what does the P-value in the table refer to? Are these peaks enriched relative to an input or to a cell line not expressing Flag? Are enhancers for known Hedgehog pathway targets (Ptch1, Gli1) also bound by Gli2?

We had up-loaded the raw data to the SRA database (BioProject: PRJNA374459) and provided the detailed methods for analyzing the ChIP-seq data in the revised Materials and methods section. We now also include a supplemental Excel file showing analyses results from both ChIP-seq and RNA-seq. The peaks enriched were relative to cells not induced with Dox and therefore not expressing Flag. We indeed see *Gli2* binding to both Ptch1 and Gli1 locus, with the latter now shown in Figure 5).

6) Besides the known roles for Wnts in inhibiting adipogenesis, it is not clear why the focus is on the three Wnt genes for the ChIP experiment. In my quick examination of the predicted target genes provided in the excel spreadsheet using an online KEGG enrichment analysis tool, the most upregulated pathways were (in order of being most statistically enriched: Jak-Stat (25 genes), Notch (13 genes), Tgf-β (16 genes), Wnt (19 genes). This cursory analysis suggests that a strong rationale for focusing exclusively on Wnt needs to be provided. A confounding factor in the current level of data analysis is that, if my understanding is correct, the nearest gene in the genome is shown on the dataset regardless of whether it is transcriptionally active in adipocytes. Could the ChIP regions be intersected with Hedgehog-responsive genes in adipocytes (or if this is unavailable at least genes known to be transcribed in adipocytes) to predict which of the ChIP regions are functional?

We thank the reviewer for the constructive comment. We have previously generated a RNA-seq dataset in M2 cells after 72 hrs of purmorphamine treatment (Shi et al., 2015), and now provide a full list of genes with a minimum of 2-fold change in expression (supplemental Excel file). By intersecting the RNA-seq and ChIP-seq data, we identified a list of candidate Hh target genes, these including three Wnt genes. Analyses of the candidate target genes with DAVID also identified Wnt signaling as a relevant pathway. We have revised the text to explain our thought process (subsection “Wnt6 is a direct target of Hh-*Gli2* signaling”).

7) Related to concern 4, I am concerned that these Wnt sites may not represent direct Gli targets. The ability the Gli-bound sites to act as enhancers should be tested in cell culture enhancer assays in the presence or absence of Hedgehog pathway activation.

We appreciate the reviewer’s concern. We acknowledge that Wnt5a and Wnt9a may not be bona fide targets in vivo as they were not induced in either SmoM2 or ΔNGli2 mice (Figure 5). We have further pursued Wnt6 as it was induced upon Hh activation both in vitro and in vivo. We tested the four Gli-bound sites individually in a cell culture assay and found that site 1 and 3 each mediated ~60% induction of luciferase expression in response to purmorphamine (Figure 5).

8) Please indicate the genetic background of the mice in the experiments.

The mice were in a mixed genetic background of C57BL/6 (~70%) and 129 (~30%).

9) Are there sex-specific differences in Hh-mediated inhibition of obesity? If so, they should be shown. If not, this should be stated.

There is no sex difference in Hh-mediated inhibition of obesity. Data from either male or female are presented as stated in figure legends.

10). Figure 9A-C: What is the time course of the Wnt inhibition experiments? Please provide appropriate details in the Materials and methods / Results sections.

These data are now presented in Figure 7. The details are now provided in the Results section.

[Editors' note: the author responses to the re-review follow.]

Essential revisions:The major concern remains the link between Hedgehog signalling and Wnt activation, since the results do not strongly support the authors' conclusions. These concerns are expanded in the specific comments below. While the data strongly suggest that Gli2 regulates Wnt activity, they do not provide compelling support the direct transcriptional regulation of Wnt proteins by Gli2.1) The enhancer experiments shown in Figure 5 are not convincing. First, the two sites are less than 2-fold activated. For in vitro GLI enhancer assays, this is not a compelling increase in activity. It is possible that this is because the M2 cells were only treated with an unspecified amount of purmorphamine for 24 hours when previous work by this lab showed maximal stimulation was achieved at 48 hours (Shi et al., 2015). Alternatively, perhaps the M2 bone marrow line does not model adipogenic responses under the current culture conditions, or that Wnt signaling is not active under these biological conditions. Finally, it is certainly still possible that these regions are not biological enhancers. Taken together, these experiments do not adequately support their conclusion that Wnt6 as a direct transcriptional target of Gli2. I can think of other approaches that could be used to test this (in vivo transgenic enhancer assays or possibly CRISPR-based deletions that are clearly outside the scope of the current work and might ultimately end up showing no enhancer activity.2) The ChIP-seq data in Figure 5 shows relatively weak enrichment of Gli2 at the Wnt6 region compared to binding at Gli1 or Wnt9a (Figure S5). This, combined with the fact that the ChIP data was acquired from a lentivirally driven Gli2 construct in M2 cells does not provide high levels of confidence that these represent Gli2 binding in adipocytes.3) Discussion section "Furthermore, Gli2 functions at least partly through direct transcriptional regulation of several Wnt proteins"; Discussion section "Further downstream, several Wnt genes are direct transcriptional targets of Gli2." Both of these statements convey the erroneous perception that Gli22 directly regulates multiple Wnt proteins. Their own data suggests that Wnt5a and 9b may not be in vivo targets (as they themselves acknowledge in subsection “Wnt6 is a direct target of Hh-Gli2 signaling”). Thus, at best, it could be claimed that Gli2 regulates the transcriptional regulation of a Wnt protein. However, the evidence that Gli2 regulates Wnt6 is unconvincing.4) The new supplemental spreadsheet showing ChIP-seq reads is helpful. However, both within the supplemental dataset and subsection “ChIP-seq experiments”, there is no statistical information on the statistical analysis done for peak calling (stated that this is done in Partek but this is not sufficient information for a reviewer or reader to know the threshold metric used for calling a peak). Similarly, in the supplemental spreadsheet, there is a 'scaled fold change'. While it is not mentioned how this was calculated, it seems like it is a modified read count in dox treated versus dox untreated. The authors should calculate a modified P-value to reflect the quality of these peaks.5) The authors need to tone down their claims of a direct transcriptional link between Shh and Wnt, and remove the data that the reviewers consider to be weak (see above).

According to your recommendation, we have toned down the conclusion about Wnt6 being a direct target gene of *Gli2*, but instead focus on Wnt signaling being induced downstream of Hh activation. Please note the major changes shown in red in the text. We deleted the enhancer experiment (previously Figure 5) that the reviewer considered to be weak (Specific Point 1), corrected the text in the Discussion section (Specific Point 3), and also provided additional bioinformatics information about the ChIP-seq analyses in the Materials and methods section (Specific Point 4).

[Editors' note: further revisions were requested prior to acceptance, as described below.]

The manuscript has been improved but there are some remaining issues that need to be addressed before acceptance, as outlined below. Specifically, the reviewers request that you remove all the ChIP-seq data from the main body of the paper and the abstract. In addition, since none of the in vitro assays appears to be adipogenic, the manuscript lacks a solid connection between the in vivo findings and the downstream molecular mechanism studies and you should address this within the text.As highlighted previously there are several problems with your ChIP-seq data: 1) there are many binding regions that do not have enhancer activity – so this binding does not necessarily imply that the regulation is direct, and 2) the ChIP-seq results were obtained in M2 cells.

According to your recommendation, we have removed all the ChIP-seq data and the related text from the main body as well as the abstract. We believe the in vitro studies provide supportive evidence for the potential involvement of Wnt activation, and therefore have decided to keep them in the revision. At the same time, we acknowledge the limitation of such studies in both Results and Discussion sections, and state clearly that the in vivo relevance of such findings needs to be tested in vivo.